# EVINCE: Optimizing Adversarial LLM Dialogues via Conditional Statistics and Information Theory

## Abstract

This paper introduces EVINCE (Entropy and Variation IN Conditional Exchanges), a framework that optimizes multi-LLM dialogues using conditional statistics and information theory. EVINCE introduces dual entropy optimization to balance perspective diversity with prior knowledge, providing quantitative measures for modulating LLM interactions. Through information-theoretic metrics and mutual information optimization, the framework demonstrates consistent improvement over single-LLM performance in applications ranging from disease diagnosis to news debiasing. We present theoretical foundations and empirical validation for this structured approach to LLM collaboration.

## 1 Introduction

Current Large Language Models (LLMs) (e.g., Anthropic (2024); OpenAI (2023); Gemini (2023)) demonstrate remarkable capabilities but face significant challenges, including hallucination (generating false information), bias (reflecting societal prejudices), and limited reasoning (difficulties in complex problem-solving and logical inference). These challenges critically limit LLMs' reliability and applicability in real-world scenarios.

Multi-agent dialogue systems offer a promising direction to address these limitations. By fostering diversity and debate among LLMs, these systems can potentially mitigate biases and enhance reasoning capabilities through structured interaction and collaborative intelligence. However, most prior works in multi-agent debate (e.g., Abdelnabi et al. (2024); Chan et al. (2023); Fu et al. (2023); Li et al. (2023); Liang et al. (2023); Michael et al. (2023); Smit et al. (2024)) merely follow traditional mixture of experts approaches Jacobs et al. (1991). These methods primarily use redundancy to mask errors Freund & Schapire (1997) and leverage error diversity to improve response quality. Their debates often lead to problems such as argument repetition, reinforcement of low-quality reasoning, or inconsistent outputs Wang et al. (2024); Zhang et al. (2024).

While traditional approaches focus on error reduction, recent work SocraSynth Chang (2023a) explored dynamic modulation of debate "contentiousness" to generate diverse perspectives and then refine with reasoning to improve results. By leveraging both adversarial and collaborative interactions between LLMs, SocraSynth demonstrates quantifiable improvements across various domains, including healthcare Chang & et al. (2023), sales planning Tsao (2023), and emotional behavior modeling Chang (2024). However, its qualitative approach to modulating contentiousness in linguistic behavior limits precise control and optimization of these interactions.

In this paper, we advance this line of research by introducing EVINCE (Entropy and Variation IN Conditional Exchanges), which provides quantitative foundations for multi-agent dialogue moderation. EVINCE builds on three theoretical pillars:

1. *Inclusiveness Exploration*: We develop methods to ensure dialogues explore all potential perspectives. We use conditional statistics to "free" an LLM agent from its default "maximum likelihood" next-token

prediction behavior, allowing it to adopt specific stances. We introduce a dual entropy optimality theory to balance the exploration of new ideas with adherence to priors, thus optimizing information exchange between agents for comprehensive and stable discourse.

2. *Information Flow Dynamics*: We introduce information theory-based metrics to quantify and optimize dialogue dynamics. These measure information diversity (entropy), novelty (statistical divergence scores), and inter-agent persuasion (mutual information). These metrics enable us to assess and enhance the quality and efficiency of information flow within the multi-agent system, fostering rich and productive exchanges.

3. *Reasoning Quality and Coherence*: We establish frameworks to assess the logical structure and coherence of multi-agent reasoning. This pillar evaluates argument validity, analytical depth, and dialogue coherence. We synergistically integrate the CRIT algorithm Chang (2023b), which combines Socratic methods with formal reasoning techniques, enhances our ability to conduct critical thinking through evaluating argument quality, information-source credibility, and overall "reasonableness" within the dialogue. This integration ensures that the collective reasoning of LLM agents is not only diverse but also logically sound and aligned with the dialogue's objectives.

The contributions of this paper are:

1. *Framework Design*: Unlike approaches using debate merely for accuracy via redundancy, EVINCE facilitates information discovery, bias mitigation, and decision-making requiring both breadth and depth of information.

2. *Theoretical Foundations*: We establish EVINCE's theoretical basis in conditional Bayesian statistics, mutual information, and dual entropy. Our novel dual entropy theory demonstrates how productive decision-making should start with diverse inputs and stable objectives, then converge through information exchange to optimal predictions.

3. *Empirical Validation*: Through studies in disease diagnosis (Section 3) and news debiasing (Appendix G), we validate EVINCE's theories and derive practical maxims for optimizing information flow and prediction accuracy.

## 2 EVINCE ALGORITHM, MAXIMS, AND THEORIES

**Problem Statement:** Organize a structured debate between two equally competent large language models (LLMs), LLM$_A$ and LLM$_B$, to conduct $t$ rounds. At each round $t$, each model generates confidence scores for its top-k predictions, denoted as $P_A^{(t)}$ and $P_B^{(t)}$, over $C$ possible outcomes, accompanied by supporting arguments $R_A^{(t)}$ and $R_B^{(t)}$. The goal is to design an iterative debate process that leverages the structured exchange of arguments to enable the models to converge on an optimal prediction ranking $P^*$ across the $C$ classes. Optimality is defined as achieving the highest possible accuracy on the prediction task while maintaining strong reasoning support.

### 2.1 PRELIMINARIES IN INFORMATION THEORY

This section summarizes the key metrics used to measure information diversity, similarity, divergence, and other relevant factors within EVINCE. These metrics serve three primary objectives:

#### 2.1.1 FOSTERING DIVERSITY OF PERSPECTIVES WHILE ENSURING REASONING QUALITY

- Wasserstein Distance (WD): Measures distribution difference between predictions to identify exploration opportunities Kantorovich (1942); Rubner et al. (2000); Villani (2008).
- Shannon Entropy or Relative Entropy: Measures diversity of perspectives Shannon (1948).

- Reasoning Quality: The CRIT (Critical Thinking) algorithm evaluates the logical soundness and persuasiveness of supporting arguments (Appendix B), helping to identify and mitigate hallucinations and poorly-reasoned arguments Chang (2023b).

---

INPUT: Information set $S$, Class labels $C$; $\text{LLM}_A$ and $\text{LLM}_B$; (**Maxim #1**)
OUTPUT: $P_f$, final top-$k$ confidence distribution over $C$ classes; $R = \emptyset$ aggregated arguments;
VARIABLES:
  $t = 0$: debate round; $R_A^{(t)}$, $R_B^{(t)}$: supporting reason sets;
  $P_A^{(t)}$, $P_B^{(t)}$: top-$k$ confidence distributions of $\text{LLM}_A$ and $\text{LLM}_B$ on $C$ of round $t$; (**Maxim #4**)
  $\Delta = 90\%$: debate contentiousness, initialize to high to foster exploration over exploitation; (**Maxim #2**)
  $p$: prompt = "Predict top-$k$ confidence distribution on $C$ with $S$ and $R$ at contentiousness level $\Delta$";
FUNCTIONS: $\Omega = \text{CRIT}()$, for evaluating argument quality; WD(), MI(), information theory metrics;

**BEGIN**

1: INITIAL ROUND:
  $\text{LLM}_A$ generate its prediction and $\text{LLM}_B$ refutes with its own prediction:
$$(P_A^{(t=0)}, R_A^{(t)}) = \text{LLM}_A(S, C, p, R); \quad (P_B^{(t=0)}, R_B^{(t)}) = \text{LLM}_B(S, C, p, P_A^{(t=0)}, R = R \cup R_A^{(t)});$$
$$\text{WD}_{old} = \text{WD}(P_A^{(0)}, P_B^{(0)}); \quad \text{MI}_{old} = \text{MI}(P_A^{(0)}, P_B^{(0)}); \quad \text{CRIT}_{old} = \text{CRIT}(S, R_A^{(0)}, R_B^{(0)});$$
2: DEBATE ITERATIONS:
  WHILE ( $\text{WD}(P_A^{(t)}, P_B^{(t)}) \leq \text{WD}_{old}$ & $\text{MI}(P_A^{(t)}, P_B^{(t)}) \geq \text{MI}_{old}$ & $\text{CRIT}(S, R_A^{(t)}, R_B^{(t)}) \geq \text{CRIT}_{old}$ )
  2.1. LLMs generate new predictions w/ arguments:
$$(P_A^{(t+1)}, R_A^{(t+1)}) = \text{LLM}_A(P_B^{(t)}, S, C, p, R = R \cup R_B^{(t)});$$
$$(P_B^{(t+1)}, R_B^{(t+1)}) = \text{LLM}_B(P_A^{(t+1)}, S, C, p, R = R \cup R_A^{(t+1)});$$
  2.2. Update contentiousness level and update parameters (**Maxim #3**)
$$\text{WD}_{old} = \text{WD}(P_A^{(t)}, P_B^{(t)}); \quad \text{MI}_{old} = \text{MI}(P_A^{(t)}, P_B^{(t)});$$
$$\text{CRIT}_{old} = \text{CRIT}(S, R_A^{(t)}, R_B^{(t)}); \quad t = t + 1; \quad \text{Update}(\Delta);$$
3: CONCILIATORY OUTPUT:
  Generate weighted prediction by quality scores $\Omega$ from CRIT ; (**Maxim #4**)
$$P_f = (\Omega_A P_A^{(t)} + \Omega_B P_B^{(t)})/\Omega_A + \Omega_B; \quad \text{RETURN}(P_f, R \cup R_B^{(t)});$$
**END**

---

Figure 1: **Specifications of Algorithm** EVINCE. Key points: 1) Asymmetric Start: In Step #1, $\text{LLM}_A$ initiates with opening arguments based solely on the given information, while $\text{LLM}_B$ starts with access to $\text{LLM}_A$'s prediction and arguments for refutation. 2) Termination Criteria: The while loop in Step #2 considers three factors: Wasserstein distance, mutual information, and argument quality. EVINCE terminates if the dialogue ceases to make significant progress. 3) Further Details: Maxims #1 to #4 provide additional explanations of the algorithm's principles. 4) Argument Evaluation: Step #2.2 evaluates of argument quality, and the while loop examines if argument quality continues to improve. 5) Update($\Delta$) modulates contentiousness, and see Maxim #3 for its specifications.

### 2.1.2 EXPLORING NEW POSSIBILITIES WHILE ADHERING TO THE DIALOGUE SUBJECT

- Correlation Coefficients: Tracks the evolution of opinions and assesses debate stability Brown et al. (2005) toward the goal of the dialogue.

- Mutual Information (MI): Quantifies information overlap to ensure focused and productive debates Cover & Thomas (2006b) and measures the degree of agreement/disagreement.

### 2.1.3 EXAMINING INFORMATION CONVERGENCE AND ESTABLISHING TERMINATION CRITERIA

- Jensen-Shannon (JS) Divergence: Assesses similarity between probability distributions (symmetric).

- Cross Entropy (CE): Measures the asymmetric difference between prediction distributions.

- Kullback-Leibler (KL) Divergence: Reveals asymmetric differences between probability distributions.

Appendix A summarizes these metrics and their pros and cons. Next, we present the EVINCE algorithm and explain how these metrics are used to moderate an LLM dialogue to achieve a balance between exploration and exploitation, leading to optimal prediction outcomes.

Figure 1 specifies the detailed operations of EVINCE. EVINCE employs two equally competent LLM instances, $LLM_A$ and $LLM_B$, which can be different models (e.g., GPT and Claude) or independent instances of the same model. Given an information set $S$ and a class-label set $C$, EVINCE outputs a confidence distribution over $C$ with justifications. For example, $S$ could represent a patient's symptoms and $C$ a set of diseases, with EVINCE moderating a dialogue to predict the patient's disease(s).

**Moderation Subroutines:** EVINCE employs four key subroutines to manage the exploration-exploitation tradeoff, ensuring information diversity, quality, and stability:

- CRIT: Evaluates argument quality and source credibility. Low scores can trigger dialogue termination.
- WD (Wasserstein distance): Assesses prediction diversity, expected to decrease over time.
- MI (Mutual Information): Evaluates dialogue convergence. Stagnation can trigger termination.
- Additional metrics: KL divergence, Jensen-Shannon divergence, and cross entropy ensure evaluation consistency (see Table 1 in Appendix A).

When all metrics plateau, indicating no further improvement, EVINCE terminates the dialogue. At this stage, both LLM instances are prompted to collaboratively deliver a conciliatory conclusion that includes comprehensive arguments and counterarguments. They also provide a list of missing information that, if obtained, could enhance prediction accuracy and reliability, potentially using Retrieval-Augmented Generation (RAG) techniques.

**Dialogue Iterations (Steps 1 and 2):** Given the contentiousness level set for a dialogue iteration, each LLM generates a new prediction distribution on $C$ with supporting arguments, considering the other LLM's previous prediction and accumulated reasoning as context. The contentiousness level updates follow Maxims #2.2 and #3. The top-k predictions from LLMs are then calibrated using reasoning quality assessment following Maxim #4.

**Final Output (Step 3):**

- Combine final predictions from both LLMs.
- Weight predictions based on supporting argument quality (evaluated by CRIT).
- Handle uncertainty by soliciting missing information when final prediction entropy is high (Maxim #4.4).

EVINCE facilitates a structured debate between AI models, unearthing all perspectives related to the prediction tasks at hand while balancing diverse viewpoints and fostering consensus to produce accurate, well-reasoned predictions.

## 2.2 MAXIMS WITH THEORETICAL FOUNDATIONS

Progress towards the optimality goal is guided and measured by metrics introduced in Section 2.1. This section explains how these metrics can be used in complementary ways to facilitate proper trade-offs between diversity and convergence, exploration and exploitation, and several other factors. In the EVINCE algorithm presented in Figure 1, we have annotated the steps to which these four maxims are applied.

**Maxim #1. Orchestrate Two Equally Competent LLMs in Structured Debate:** Integrating two equally competent LLMs ensures a balanced exchange of insights and avoids bias. This adversarial setup fosters diversity in predictions, each supported by justifications, promoting critical evaluation and uncovering potential blind spots.

*Methods*: Choosing LLMs with comparable performance on a shared validation set, a balanced debate can be ensured. Suitable models (in 2024) include GPT-4, Claude, and Gemini. Conditioning different instances of the same LLM to support opposing stances on a subject matter can also be effective due to the theoretical justification of in-context learning with conditional Bayesian statistics Xie et al. (2021).

**Maxim #2. Fostering Exploration through Diverse Perspectives:** While LLMs default to maximum likelihood predictions favoring popular outcomes, context conditioning can shift them toward exploration over exploitation. High initial contentiousness encourages dynamic debate and challenges to prevailing views, mitigating confirmation bias through contrary queries and diverse top-k predictions.

*Methods*: Set high initial contentiousness to stimulate exploration, enabling LLMs to challenge each other's predictions. Monitor mutual information and agreement diversity to ensure productive debate.

**Maxim #3. Refining High-Quality Perspectives:** Once new insights plateau (measured by mutual information, Wasserstein distance, and KL divergence), shift from exploration to exploitation by reducing contentiousness.

*Methods*: Algorithm Update($\Delta$) guides this transition to a conciliatory stage, focusing on strengthening well-supported arguments for the final output.

**Maxim #4. Combine Predictions Weighted by Diversity and Quality:** Combine the probability distributions from two LLMs by weighting them based on distributional diversity and argument quality. While LLMs face fundamental challenges in generating accurate probabilities (Peng & Zhang (2024); Scholten et al. (2024)), we can use reasoning-calibrated confidence scores as an effective proxy for weighting predictions.

*Methods*: Following these four maxims:

- **Maxim #4.1 Prediction Reliability:** Use entropy-based measures to estimate reliability. Lower entropy suggests higher confidence and greater reliability in predictions.
- **Maxim #4.2 Argument Quality:** Evaluate argument quality using CRIT, identifying logical fallacies and assessing the relevance and credibility of evidence.
- **Maxim #4.3 Aggregation:** Apply a weighted aggregation method, such as a Bayesian model, to combine predictions, accounting for both probabilistic insights and argument quality.
- **Maxim #4.4 Diagnosis and RAG:** If the final prediction has high entropy, indicating uncertainty, diagnose the issue and perform Retrieval-Augmented Generation (RAG) to obtain missing information.

### 2.3 ENTROPY DUALITY THEOREM (EDT)

**Theorem EDT: Optimal Pairing of LLMs for Probabilistic Prediction Accuracy.** The optimal pairing of LLMs for diagnosis accuracy, in terms of stability, accuracy, and robustness, occurs when the LLMs are 1) equivalent in the quality of the information they process, and 2) exhibit contrasting entropy values in their prediction distributions—one high and one low.

*Proof.* Please see Appendix C. □

## 3 EMPIRICAL STUDY

This study examines EVINCE's application to disease diagnosis, evaluating LLMs as diagnostic tools. (A complementary study on news debiasing is documented in Appendix G.) We validate three hypotheses:

1. *Contentiousness & Prediction Quality*: Initial LLM disagreement (measured by Wasserstein distance) increases with contentiousness but converges as debate progresses. Individual LLM prediction uncertainty (Shannon entropy) follows a similar pattern.

2. *EDT Effectiveness*: LLM pairs following the Entropy Duality Theorem (EDT) exhibit complementary error patterns, achieving higher combined prediction accuracy than non-EDT pairs.

3. *Misdiagnosis Detection*: Applied to real-world data, EVINCE improves diagnostic accuracy while identifying potential misdiagnoses and ambiguities in ground truth labels.

**Problem Statement:**  Given a set of symptoms $S$ and a context $\kappa$, the goal is to generate and rank top-k disease predictions from a set of C possible diseases. This is represented as $P = \text{LLM}(S, \kappa)$, where each LLM generates confidence scores for its top-k predictions ($k \leq |C|$) based on the input symptoms $S$ and context $\kappa$.

$$P = (p(\text{top } 1 \text{ to } k \in C \mid S, \kappa)) . \tag{1}$$

Context $\kappa$ allows dual entropy adjustment through three parameters: temperature, top-$k$ value, and contentiousness level $\Delta$. A distribution tends toward high entropy when these parameters are set high, and toward low entropy when set low.

**Note:** Maxim #4 states that the initial confidence distribution $P$ from LLMs (Eq.1) represents confidence scores rather than absolute probabilities. In the EVINCE framework, these scores are calibrated through multi-iteration debates where predictions must be supported by arguments and examined by counterarguments. The CRIT algorithm (detailed in Appendix B) evaluates reasoning quality to adjust these confidence scores—reducing credibility when high-confidence predictions lack strong supporting evidence.

Since EVINCE and CRIT can employ different LLMs, and CRIT focuses solely on evaluating reasoning quality via Socratic methods, this collaborative approach uses reasoning quality as a proxy for confidence calibration. This sidesteps the fundamental challenge of generating reliable absolute probabilities from LLMs (Peng & Zhang (2024); Scholten et al. (2024)).

For example, if an LLM outputs confidence scores $P$ = (0.5, 0.3, 0.2) for its top-3 predicted diseases, and CRIT evaluates their supporting arguments as having strength (50%, 100%, 50%), the calibrated scores become (0.25, 0.3, 0.1). After normalization, the final distribution is $P$ = (0.38, 0.46, 0.16).

**Resources & Dataset:**  We use a Kaggle dataset Patil (2020) of 4,921 patient records, each containing a diagnosis and up to 17 symptoms. After deduplication, 304 unique cases across 40 diseases remain (dataset provided as supplement). We evaluate GPT-4, Gemini, and Claude-3 using their inherent knowledge, without few-shot training. Computing resources provided by Azure research grant.

**Evaluation:**  We use top-k Mean Reciprocal Rank (MRR): if a top-k prediction matches ground truth, the score is the reciprocal of its rank (1 for first, 1/2 for second, etc.); 0 if no match.

## 3.1 STUDY #1: EVINCE VS. BASELINE DIAGNOSIS ACCURACY

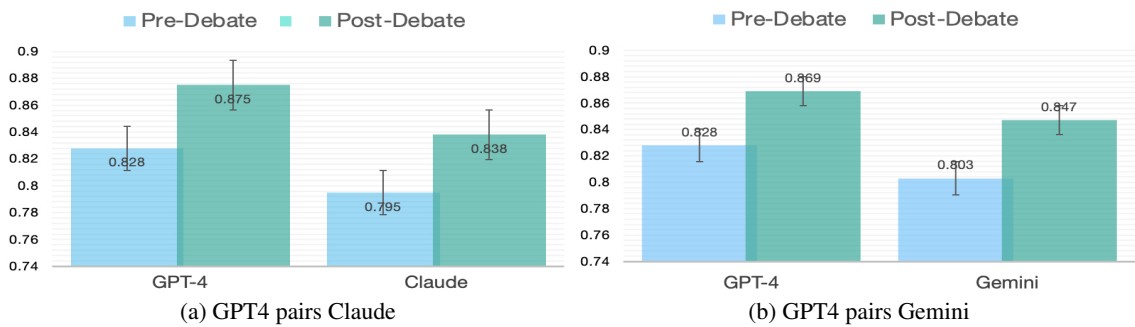

(a) GPT4 pairs Claude  (b) GPT4 pairs Gemini

Figure 2: EVINCE improves diagnosis accuracy markedly.

We evaluated EVINCE through pairwise collaborations of GPT-4, Gemini, and Claude-3 on 304 patient cases.

**Parameter Settings:** Each LLM pair uses $k = 5$ predictions with contrasting temperatures (high/low) and high contentiousness ($\Delta = 0.9$). This ensures meaningful interaction while maximizing cross entropy for information exchange.

**Evaluations:** We conducted two types of experiments:

1. **Constrained Prediction (Baseline):** Limiting predictions to the dataset's 40 disease labels yielded high accuracy (95-97%). While unrealistic for general practice where physicians consider all possibilities, this constraint demonstrates LLMs' flexibility in avoiding overfitting to potentially erroneous labels.

2. **Unconstrained Prediction:** Removing label constraints to simulate real-world conditions, all 304 cases showed consistent results across LLMs (standard deviation = 1.5%). Individual pre-debate accuracies (Figure 2, light blue bars) were: GPT-4 (82.8%), Gemini (80.3%), and Claude (79.5%).

EVINCE **Performance:** EVINCE pairings (GPT-4/Claude-3 and GPT-4/Gemini-3) improved accuracy by 4-5 percentage points (Figure 2, green bars). The GPT-4/Claude-3 pairing achieved 87.5% accuracy (Figure 2a), comparable to state-of-the-art clinical algorithms like REFUEL Peng et al. (2018).

**Discussion:** While EVINCE significantly outperforms single LLM predictions, the remaining 12.5% inaccuracy merits deeper analysis. Given the reported 11% US misdiagnosis rate Newman-Toker et al. (2023b), some discrepancies might reflect errors in the original dataset rather than EVINCE limitations. This observation suggests EVINCE's potential for identifying and correcting dataset errors, which we explore in Section 3.3.

## 3.2 STUDY #2: EXPLOITATIVE VS. EXPLOITATIVE

|        | Hep. A | Hep. B | Hep. C | Hep. D | Hep. E |
|--------|--------|--------|--------|--------|--------|
| Hep. A | 50%    |        |        |        | 50%    |
| Hep. B |        | 50%    | 50%    |        |        |
| Hep. C | 100%   |        |        |        |        |
| Hep. D |        |        |        |        | 100%   |
| Hep. E |        |        |        |        | 100%   |

|        | Hep. A | Hep. B | Hep. C | Hep. D | Hep. E |
|--------|--------|--------|--------|--------|--------|
| Hep. A | 74%    |        | 36%    |        |        |
| Hep. B |        | 50%    | 50%    |        |        |
| Hep. C |        |        | 36%    | 64%    |        |
| Hep. D | 60%    |        |        | 40%    |        |
| Hep. E |        |        |        |        | 100%   |

(a) GPT liver c-matrix          (b) Claude liver c-matrix

Figure 3: Confusion matrices

**Analysis of Confusion Matrices:** The confusion matrices for Hepatitis types A-E diagnosis (Fig. 3) reveal:

- GPT-4: Limited accuracy (50%) for types A/B, poor performance on C/D.
- Claude: Broader distribution of predictions across all types.

Claude's higher entropy predictions complement GPT-4's more confident but potentially incorrect assessments. This entropy differential drives productive debate in EVINCE, balancing exploration with exploitation for improved diagnostic accuracy.

Two primary factors contribute to EVINCE's improved diagnostic accuracy. First, structured debates with reasoning encourage LLMs to explore alternative diagnoses both broadly and deeply, leading to more comprehensive analysis and decision-making (see Appendices D and E). Second, pairing LLMs with high- and low-entropy prediction distributions, or those with a large Wasserstein distance (WD) between them,

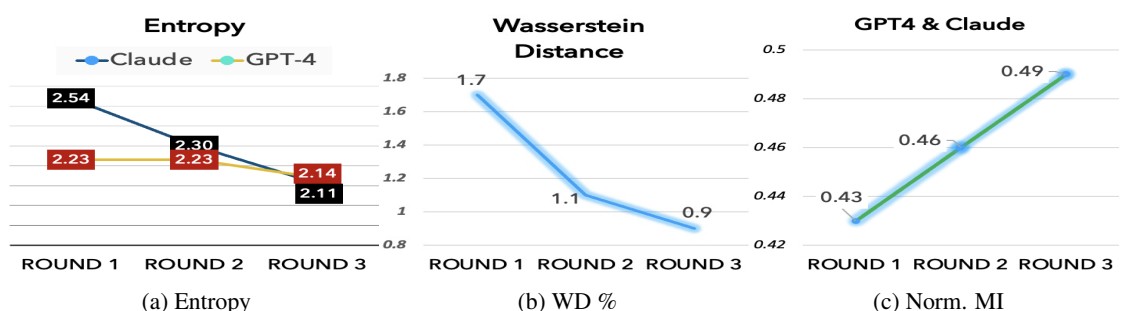

Figure 4: Entropy, WD, and normalized MI

balances exploratory diversity with exploitative stability. This approach results in more robust and higher-quality decisions, as demonstrated in this second study.

**Observations from Information Metrics:**

- Entropy Stabilization: Figure 4a shows entropy levels of both LLMs stabilizing after three debate rounds, indicating convergence towards a similar, stable entropy state.
- Wasserstein Distance Improvement: Figure 4b demonstrates consistent improvement in Wasserstein distance (WD) between the two models' predictions over successive rounds.
- Mutual Information Increase: Figure 4c reveals a 14% improvement in normalized mutual information (MI) between GPT-4 and Claude's prediction distributions, suggesting increased shared information throughout the debate.
- Convergence of Divergence Metrics: Figure 5(a) shows consistent convergence of all divergence metrics.

**Comparative Performance:** EVINCE demonstrates a 5% higher accuracy rate in diagnosing specific types of liver diseases compared to a baseline approach (Figure 2a), underscoring its capability to handle complex diagnostic scenarios effectively.

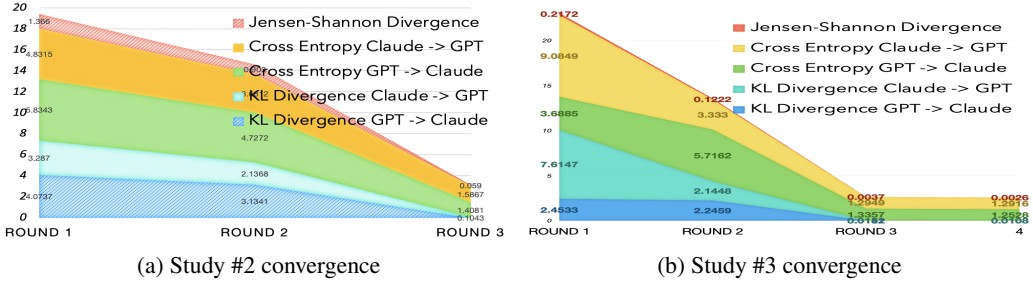

Figure 5: Convergence of all information metrics.

### 3.3 STUDY #3: EXPLAINABILITY AND GROUND-TRUTH REMEDIATION

This study illustrates how EVINCE can identify potential misdiagnoses, explain the reasoning behind them, and recommend corrective actions. Traditionally, machine learning scientists rely on labeled data as "ground truth." However, as evidenced by research like that of Newman-Toker et al. (2023a) from Johns Hopkins, misdiagnosis is a widespread issue in healthcare systems globally. These erroneous diagnoses, often treated as ground truth, can be perpetuated by supervised learning algorithms, exacerbating the problem within the healthcare system. EVINCE's dialogue capabilities provide insights into the decision-making process

and highlight missing information, helping to rectify erroneous predictions and redefine the ground truth. This approach offers a potential solution to the compounding effect of misdiagnoses in machine learning applications in healthcare.

Figure 6, which plots the respective entropies of GPT-4 and Claude, reveals two key insights. First, an large gap in Wasserstein Distance (WD) exists between the two models in the initial rounds. This disparity underscores the role of dual entropy in fostering information exchange. As the entropy values converge in rounds 3 and 4 and WD decreases significantly, we observe a corresponding convergence and stabilization of their mutual information. The information metrics that EVINCE employs effectively show the progress of the dialogue and convergence from explorative to consensus. Figure 5(b) illustrates the convergence of all divergence metrics—including Jensen-Shannon divergence, cross-entropy, and Kullback-Leibler divergence—particularly between the second and third rounds.

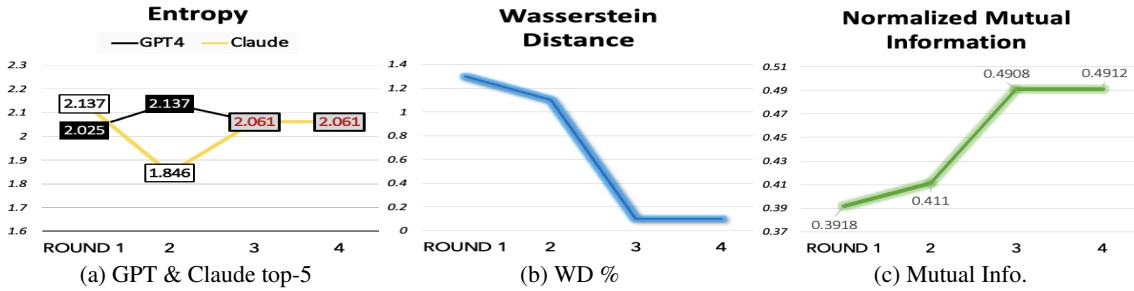

Figure 6: Remediation: Jaundice to Hepatitis

Appendix F demonstrates that EVINCE recommends additional symptoms to inquire about with the patient, as well as lab tests to confirm the diagnosis. These suggestions, validated by our hospital partners, provide valuable information to enhance diagnostic accuracy and correct errors.

## 4    CONCLUDING REMARKS

The power of EVINCE lies not only in its enhanced accuracy but also in its ability to elucidate the decision-making process and identify missing information, providing critical insights for correcting errors.

The core strength of EVINCE is built on three theoretical pillars: Inclusive Exploration, Information Flow Dynamics, and Reasoning Quality and Coherence. Grounded in conditional statistics and information theory, these pillars enable LLMs to go beyond conventional "maximum likelihood" behaviors, exhibiting human-like iterative adaptability in linguistic tasks. The integration of the CRIT system, which combines Socratic methods with formal reasoning techniques, further strengthens critical thinking and ensures logically sound, goal-oriented collective reasoning.

EVINCE leverages the advanced reasoning capabilities of post-GPT-4 LLMs to balance the exploration of diverse perspectives with the exploitation of existing knowledge, thereby maximizing their potential. Some may question how combining two LLMs, which are prone to hallucinations, can reduce those very hallucinations. EVINCE accomplishes this by effectively enhancing contextual understanding and improving the quality of inter-LLM queries, surpassing the capabilities of traditional single-round LLM interactions.

Our research demonstrates that EVINCE makes significant strides towards achieving AGI by enhancing three core characteristics: versatility, iterative adaptability, and reasoning capability. By addressing critical limitations of current AI systems—such as hallucinations, bias, and restricted reasoning abilities—EVINCE offers a promising pathway towards the development of more robust and capable artificial intelligence.

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

## APPENDIX A: FORMULAS OF METRICS AND COMPARISONS

This appendix outlines the mathematical formulas for various data analysis metrics used in probabilistic and statistical modeling. Table 1 compares their pros and cons.

### KULLBACK-LEIBLER DIVERGENCE

The Kullback-Leibler Divergence measures the difference between two probability distributions:

$$D_{KL}(P\|Q) = \sum_{x \in \mathcal{X}} P(x) \log\left(\frac{P(x)}{Q(x)}\right).$$

### JENSEN-SHANNON DIVERGENCE

The Jensen-Shannon Divergence is a symmetrized and smoothed version of the KL Divergence:

$$JSD(P\|Q) = \frac{1}{2}D_{KL}(P\|M) + \frac{1}{2}D_{KL}(Q\|M)$$

where $M = \frac{1}{2}(P + Q)$.

### WASSERSTEIN DISTANCE

The Wasserstein Distance, also known as the Earth Mover's Distance (EMD), measures the distance between two probability distributions:

$$W(P,Q) = \inf_{\gamma \in \Gamma(P,Q)} \int_{\mathcal{X} \times \mathcal{Y}} d(x,y) \, d\gamma(x,y).$$

| Metric | Pros | Cons | Remedies |
|---|---|---|---|
| Cross Entropy (CE) Shore & Johnson (1980) | Measures how well the predictions of a model fit the actual distribution of another one's outputs; asymmetric. | Computationally intensive especially with large models and data sets; sensitive to the exact nature of probability distributions. | Optimize computation strategies; use approximations or sampling methods to manage large data sets or complex models. |
| Entropy Shannon (1948) | Indicates level of diversity; high suggests exploration of possibilities, and low for confidence on few choices | High entropy might indicate noise rather than useful diversity; low entropy might mask important variability. | Use critical reading methods (CRIT) to assess argument quality; implement noise detection to differentiate between useful diversity and noise. |
| Jensen-Shannon Div. (JS) Lin (1991) | Symmetric and bounded (0 to 1), providing an interpretable measure of distributional differences. | May be less sensitive to small differences between distributions. | Increase sensitivity settings or resolution of the metric; combine with other metrics to capture finer distinctions between distributions. |
| KL Divergence Kullback (1951) | Measures difference between two probabilistic distributions. | Asymmetric; not well-defined if a distribution has zero probabilities | Use smoothing techniques to avoid zero probabilities; consider symmetric alternatives like JS divergence |
| Mutual Info (MI) Shore & Johnson (1980) | Measures reduction of uncertainty; symmetric. | Does not indicate the directionality of information flow. | Supplement with directional information metrics; normalized with max entropy of A and B. |
| Wasserstein Distance (WD) Kantorovich (1942) | Direct measure of how similar or different the model outputs are; it depicts symmetric relationship. | Not bounded but can be normalized or bounded for consistent interpretation. | Define context-specific bounds for low, medium, and high divergence; consider normalization for non-directional comps. |

Table 1: Summary of metrics for assessing LLM debates.

CROSS ENTROPY

Cross Entropy measures the average number of bits required to identify an event from a set of possibilities, under a specific model:

$$H(P, Q) = -\sum_{x \in \mathcal{X}} P(x) \log(Q(x)).$$

MUTUAL INFORMATION

Mutual Information measures the amount of information that one random variable contains about another random variable:

$$I(X; Y) = \sum_{y \in \mathcal{Y}} \sum_{x \in \mathcal{X}} p(x, y) \log \left( \frac{p(x, y)}{p(x)p(y)} \right).$$

NORMALIZED MUTUAL INFORMATION

Normalized Mutual Information is calculated as the mutual information divided by the maximum of the entropies of the variables:

$$NMI(X; Y) = \frac{I(X; Y)}{\max(H(X), H(Y))}.$$

```
         | Function Γ = CRIT(d)
         +─────────────────────────────────────────────────
         | Input. d: document; Output. Γ: validation score;
         | Vars. Ω: claim; R & R′: reason & counter reason set;
         | Subroutines. Claim(), FindDoc(), Validate();
         | Begin
    #1   |   Identify in d the claim statement Ω;
    #2   |   Find a set of supporting reasons R to Ω;
    #3   |   For r ∈ R eval r ⇒ Ω
         |       If Claim(r), (γr, θr) = CRIT(FindDoc(r));
         |       else, (γr, θr) = V(r ⇒ Ω);
    #4   |   Find a set of rival reasons R′ to Ω;
    #5   |   For r′ ∈ R′, (γr′, θr′) = V(r′ ⇒ Ω) eval rival arguments;
    #6   |   Compute weighted sum Γ, with γr, θr, γr′, θr′.
    #7   |   Analyze the arguments to arrive at the Γ score.
    #8   |   Reflect on and synthesize CRIT in other contexts.
         | End
```

Table 2: CRIT Pseudo-code. (The symbol $\Rightarrow$ denotes both inductive and deductive reasoning.)

## APPENDIX B: CRIT, CRITICAL THINKING, EVALUATIVE PHASE OF EVINCE

EVINCE uses the Socratic method to evaluate the "reasonableness" of a set of arguments that support a subject matter. The Socratic method is a questioning technique used in teaching and philosophy to encourage critical thinking and self-discovery Wikipedia (2023). The method involves asking a series of questions to explore complex ideas and help individuals arrive at their own understanding of a concept. It is based on the belief that knowledge cannot be simply imparted, but must be discovered through a process of questioning and dialogue.

To illustrate how these methods can practically be applied, let's use the example of critical reading. Critical reading is a crucial component of critical thinking, which involves evaluating the quality and credibility of written materials, from research papers to blog posts Lai et al. (2017); Paul & Binker (1990). It requires a systematic and analytical approach, asking relevant questions, and using effective prompts to gain deeper understanding of the text Elder & Paul (2010).

To aid in critical reading, we introduce a prompt template called CRIT Chang (2023b), which stands for Critical Reading Inquisitive Template. Given a document $d$, CRIT evaluates it and produces a validation score $\Gamma$. Let $\Omega$ denote the conclusion or claim of $d$, and let $R$ be the set of reasons supporting the claim. We define $(\gamma_r, \theta_r) = V(r \Rightarrow \Omega)$ as the causal validation function, where $\gamma_r$ denotes the validation score, $\theta_r$ the source credibility score, for each reason-to-conclusion argument $r \Rightarrow \Omega$. Table 2 presents the pseudo-code of $\Gamma = CRIT(d)$, which generates the final validation score $\Gamma$ for document $d$ with justifications.

EVINCE uses CRIT to evaluate argument quality of the participating LLMs involved in the debate. The input to CRIT from each LLM is first its stance on the debate subject, e.g., a set of predicted diseases, and the arguments are its reasons to arrive at the prediction. Each document in the case of EVINCE is the prediction set as the conclusion $\Omega$, the arguments as set $R$, and the opposing LLM's counterarguments as $R'$. With this document, CRIT is able to produce validity and credibility scores in $\Gamma$ for the LLM.

For detailed prompts, examples, and an empirical study verifying the effectiveness of CRIT, please consult Chang (2023b).

## APPENDIX C: PROOF OF EDT THEOREM

**Theorem EDT: Optimal Pairing of LLMs for Probabilistic Prediction Accuracy.** The optimal pairing of LLMs for diagnosis accuracy, in terms of stability, accuracy, and robustness, occurs when the LLMs are equivalent in the quality of the information they process, and exhibiting contrasting entropy values in their prediction distributions—one high and one low.

**[Proof]:** Given two LLMs, $LLM_A$ and $LLM_B$, following Maxim #1 with prediction distributions $P_A$ and $P_B$, respectively. The information entropy of $LLM_A$, $H(P_A)$, is high, and of $LLM_B$, $H(P_B)$, is low.

**Step 1: Define the combined prediction distribution.** Let the combined prediction distribution of $LLM_A$ and $LLM_B$ be denoted as $P_C$. We can express $P_C$ as a weighted average of $P_A$ and $P_B$:

$$P_C = \alpha P_A + (1 - \alpha)P_B, \quad \text{where } 0 \leq \alpha \leq 1 \text{ and}$$
$$\alpha \text{ is decided by CRIT in Appendix A.}$$

**Step 2: Express the information entropy of the combined prediction distribution.** Using the definition of information entropy, we calculate:

$$H(P_C) = -\sum P_C(x_i) \log_2 P_C(x_i)$$
$$= -\sum_i [\alpha P_A(x_i) + (1 - \alpha)P_B(x_i)] \log_2 [\alpha P_A(x_i) + (1 - \alpha)P_B(x_i)].$$

**Step 3: Apply Jensen's Inequality to the information entropy of the combined prediction distribution.** Jensen's inequality is applied to the convex function $f(x) = -x \log_2 x$. For a convex function and a set of probabilities $p_i$, Jensen's inequality states that:

$$f\left(\sum_i p_i x_i\right) \leq \sum_i p_i f(x_i)$$

Thus, the entropy of the combined distribution is:

$$H(P_C) \geq \alpha H(P_A) + (1 - \alpha)H(P_B)$$

where equality holds when $P_A = P_B$.

**Step 4: Analyze the lower bound of the combined information entropy.** As $H(P_A)$ is high and $H(P_B)$ is low, we can express their relationship as:

$$H(P_A) = H(P_B) + \Delta, \quad \text{where } \Delta > 0.$$

Substituting this into the inequality from Step 3, we have:

$$H(P_C) \geq \alpha[H(P_B) + \Delta] + (1 - \alpha)H(P_B) = H(P_B) + \alpha\Delta.$$

**Step 5: Interpret the lower bound of the combined information entropy.** The lower bound of $H(P_C)$, and hence the robustness of the model, is maximized when $\alpha$ is maximized, which corresponds to giving more weight to the high-entropy model ($LLM_A$). This setup facilitates the exploration of diverse possibilities and enhances robustness against noise and perturbations in the input data, while still ensuring that predictions are grounded by the more certain outcomes predicted by the low-entropy model ($LLM_B$).

**Step 6: Conclude the proof.** By combining the prediction distributions of $LLM_A$ and $LLM_B$, with one having high information entropy and the other low, we achieve an optimal balance that maximizes the lower bound of the combined information entropy. This balance between exploration (high entropy) and exploitation (low entropy) optimizes overall prediction accuracy and robustness, as proved through information theory and the properties of entropy. Thus, the theorem is established.

## APPENDIX D: EVINCE DEBATE - DENGUE FEVER VS. CHIKUNGUNYA

This experiment addresses a diagnostic challenge involving multiple potential diseases. A patient presents with the following 12 symptoms: [*skin rash, joint pain, vomiting, fatigue, high fever, headache, nausea, loss of appetite, pain behind the eyes, back pain, malaise, muscle pain, red spots over the body*]. In this case, GPT-4 provides the initial diagnosis, with Gemini following suit. The confirmed diagnosis for this experiment is Dengue Fever. Given the satisfactory diversity in predictions from the two LLMs, the debate progresses and converges smoothly.

### D.1 MODERATOR'S PROMPT

"Identify potential diseases linked to the symptoms (skin rash, joint pain, vomiting, etc.). Provide top-3 predictions with justifications, supplementary symptom inquiries, and relevant lab tests."

### D.2 GPT-4'S OPENING ROUND

**Top-3 Predictions:** *Dengue Fever (60%), Chikungunya (25%), and Zika Virus (15%).*

**Justifications**: Most symptoms are classic signs of Dengue Fever. Red spots over the body could correspond to petechiae, often seen in Dengue due to thrombocytopenia (a decrease in platelet count).

Chikungunya can cause similar symptoms to Dengue, such as fever, severe joint pain, muscle pain, headache, nausea, and rash. Joint pain from Chikungunya is often very debilitating and can be more prolonged than in Dengue.

Zika Virus symptoms can include fever, rash, joint pain, malaise, and conjunctivitis, with a milder presentation compared to Dengue or Chikungunya. The presence of a rash and conjunctivitis is notable, although Zika is less likely to cause severe muscle pain or high fever.

### D.3 GEMINI'S OPENING ROUND

**Top-3 Predictions:** *Viral Infection (60%), Autoimmune Disease (20%), Bacterial Infection (15%).*

**Justifications**: Symptoms common in viral infections. Autoimmune diseases is possible, but less likely due to vomiting and nausea. Bacterial infection is less probable given rash and muscle pain.

### D.4 GPT-4'S SECOND ROUND

**Top-3 Predictions:** *Dengue Fever (60%), Chikungunya (25%), and Zika Virus (15%).*

**Justifications**: GPT-4 maintains its predictions, countering Gemini's assessment with three points. First, while viral infections can cause these symptoms, the specific combination suggests a more targeted diagnosis. Second, the acute onset of high fever, joint pain, and rash is less typical for autoimmune diseases. Finally, the rash and red spots are not characteristic of bacterial meningitis.

### D.5 GEMINI'S SECOND ROUND

**Top-3 Predictions:** *Dengue Fever (50%), Chikungunya (30%), and Viral Infection (20%).*

**Justifications**: Gemini's initial prediction considers a Viral Infection as the most likely cause due to its broad range of potential symptoms. However, GPT-4's focus on Dengue Fever and Chikungunya, with their specific

symptom profiles, provides a more precise explanation for the patient's condition. Here's what is lacking in the first set of predictions:

- *Specificity*: While a Viral Infection could explain most symptoms, it does not account for joint pain and the presence of a rash, which are more characteristic of DF or Chikungunya.
- *Detailed justifications*: The first analysis does not delve as deeply into the reasons behind each disease possibility. GPT-4's explanation for Dengue Fever and Chikungunya is more comprehensive, strengthening the connection between symptoms and diagnosis.
- *Importance of additional information*: Emphasize how geographic location and travel history can significantly impact the likelihood of these specific diseases.

### D.6 GPT-4's Third Round

**Top-3 Predictions:** *Dengue Fever (60%), Chikungunya (35%), and Zika Virus (5%).*

**Justifications**: Though GPT-4 slightly adjusts probabilities, it considers the adjustment arbitrary without new information that would specifically shift the likelihood between Dengue and Chikungunya. GPT-4 refutes Gemini's Viral Infection prediction as lacking specificity. The symptoms described are quite distinctive and align closely with specific viral infections transmitted by mosquitoes.

### D.7 Gemini's Third Round

**Top-3 Predictions:** *Dengue Fever (60%), Chikungunya (35%), and Zika Virus (5%).*

**Justifications**: Agreeing that GPT's critique is spot-on, and reiterating the need for more info.

### D.8 Final Joint Recommendations to GP

After seeking further input from GPT-4, Gemini fully concurs with GPT-4's final suggestions, even though the moderator instructs Gemini to maintain its original stance. Both LLMs jointly recommend additional symptom examinations and lab tests:

A comprehensive diagnostic approach for Dengue Fever requires a combination of clinical assessment, laboratory testing, and consideration of the patient's travel and vaccination history, as well as the epidemiological context. Clinical symptoms such as skin rash, joint pain, vomiting, and red spots over the body provide valuable clues to the diagnosis, but additional symptoms like nausea, vomiting, diarrhea, bleeding, bruising, and drowsiness can indicate the severity and progression of the disease.

**Recommended Laboratory Tests:**

- *Complete Blood Count (CBC)*: Identify thrombocytopenia or leukopenia.
- *Serology Tests*: Detect specific IgM and IgG antibodies for Dengue, Chikungunya, and Zika.
- *Polymerase Chain Reaction (PCR)*: Direct detection of viral RNA.
- *NS1 Antigen Test for Dengue*: Early and specific detection of Dengue virus.
- *Urine Test for Zika*: Viral presence in urine.

## APPENDIX E: EVINCE DEBATE - JAUNDICE VS. HEPATITIS

In this experiment, GPT-4 and Claude receive an identical list of symptoms from the moderator, aimed at guiding disease diagnosis, facilitating debate, and shaping subsequent recommendations. The "ground-truth" disease is Jaundice. (We do not shorten the debate content for review.)

### E.1 MODERATOR'S OPENING PROMPT

"What potential disease could be associated with the following symptoms: itching, vomiting, fatigue, weight loss, high fever, yellowish skin, dark urine, and abdominal pain? Please offer top-five predictions with probabilities normalized to one, supported by justifications. Additionally, in the end of debate, include a list of supplementary symptom inquiries and recommend relevant lab tests to strengthen the confidence in your prediction."

### E.2 GPT-4 OPENING ROUND

- *Hepatitis C (HCV)*: 40% **Justification**: Hepatitis C is a viral infection that can cause both acute and chronic hepatitis. The combination of symptoms like fatigue, abdominal pain, nausea, vomiting, jaundice, dark urine, and weight loss strongly suggests HCV. Itching may also occur due to liver dysfunction.
- *Hepatitis B (HBV)*: 30% **Justification**: Hepatitis B is a viral infection that can cause acute or chronic liver inflammation. Symptoms are similar to HCV, including fatigue, abdominal pain, nausea, vomiting, dark urine, jaundice, and itching. The presence of these symptoms warrants testing for both HBV and HCV.
- *Cirrhosis*: 15% **Justification**: Cirrhosis is a late stage of liver scarring that can result from various liver diseases. The combination of symptoms like fatigue, weight loss, abdominal pain, jaundice, dark urine, and itching suggests advanced liver disease, possibly cirrhosis.
- *Obstructive Jaundice*: 10% **Justification**: Obstructive jaundice occurs when there is a blockage in the bile ducts. The presence of jaundice, dark urine, itching, abdominal pain, nausea, vomiting, and weight loss suggests a possible obstruction. However, the absence of fever makes an infectious cause like cholangitis less likely.
- *Acute Liver Failure*: 5% **Justification**: Acute liver failure is a rare but life-threatening condition characterized by rapid deterioration of liver function. The presence of jaundice, abdominal pain, nausea, vomiting, fatigue, and itching, along with the acute onset of symptoms, may indicate acute liver failure. However, the absence of altered mental status (hepatic encephalopathy) makes this less likely.

### E.3 CLAUDE OPENING ROUND

- *Hepatitis B (HBV)*: 35% **Reasoning**: Hepatitis B is a viral infection that can cause acute or chronic liver inflammation. Symptoms include fatigue, abdominal pain, nausea, vomiting, dark urine, jaundice, joint pain, and itching.
- *Hepatitis C (HCV)*: 25% **Reasoning**: Hepatitis C is a viral infection that can cause both acute and chronic hepatitis. Symptoms are similar to HBV but may also include weight loss and fever. Jaundice is a common symptom.
- *Obstructive Jaundice (due to gallstones, pancreatic cancer, or bile duct obstruction)*: 20% **Reasoning**: Obstructive jaundice occurs when there is a blockage in the bile ducts, preventing bile from reaching the intestines. This can lead to yellowing of the skin and eyes, dark urine, itching, abdominal pain, nausea, and vomiting.
- *Alcoholic Hepatitis*: 15% **Reasoning**: Alcoholic hepatitis is liver inflammation caused by excessive alcohol consumption. Symptoms include abdominal pain, nausea, vomiting, fatigue, dark urine, and possibly fever and weight loss.
- *Hepatitis A (HAV)*: 5% **Reasoning**: Hepatitis A is an acute, self-limiting liver infection transmitted through contaminated food or water. Symptoms include fatigue, abdominal pain, nausea, vomiting, dark urine, and jaundice.

### E.4 GPT-4 SECOND ROUND

- *Hepatitis B (HBV)*: 35% **Symptoms**: fatigue, abdominal pain, nausea, vomiting, dark urine, jaundice (yellowish skin), joint pain, itching

- *Hepatitis C (HCV)*: 25% **Symptoms**: fatigue, abdominal pain, nausea, vomiting, dark urine, jaundice (yellowish skin), weight loss, fever, itching
- *Obstructive Jaundice (due to gallstones, pancreatic cancer, or bile duct obstruction)*: 20% **Symptoms**: yellowish skin (jaundice), dark urine, itching, abdominal pain, nausea, et al.
- *Alcoholic Hepatitis*: 15% **Symptoms**: abdominal pain, nausea, vomiting, fatigue, jaundice (yellowish skin), dark urine, fever, weight loss
- *Hepatitis A (HAV)*: 5% **Symptoms**: fatigue, abdominal pain, nausea, vomiting, dark urine, jaundice (yellowish skin)

### E.5 CLAUDE SECOND ROUND

- *Hepatitis C (HCV)*: 40% **Reasoning**: Hepatitis C is a viral infection that can cause both acute and chronic hepatitis. The combination of symptoms like fatigue, abdominal pain, nausea, vomiting, jaundice, dark urine, and weight loss strongly suggests HCV. Itching may also occur due to liver dysfunction.
- *Hepatitis B (HBV)*: 30% **Reasoning**: Hepatitis B is a viral infection that can cause acute or chronic liver inflammation. Symptoms are similar to HCV, including fatigue, abdominal pain, nausea, vomiting, dark urine, jaundice, and itching. The presence of these symptoms warrants testing for both HBV and HCV.
- *Cirrhosis (due to various causes such as chronic hepatitis, alcohol abuse, or non-alcoholic fatty liver disease)*: 15% **Reasoning**: Cirrhosis is a late stage of liver scarring that can result from various liver diseases. The combination of symptoms like fatigue, weight loss, abdominal pain, jaundice, dark urine, and itching suggests advanced liver disease, possibly cirrhosis.
- *Obstructive Jaundice (due to gallstones, pancreatic cancer, or bile duct obstruction)*: 10% **Reasoning**: Obstructive jaundice occurs when there is a blockage in the bile ducts. The presence of jaundice, dark urine, itching, abdominal pain, nausea, vomiting, and weight loss suggests a possible obstruction. However, the absence of fever makes an infectious cause like cholangitis less likely.
- *Acute Liver Failure (due to various causes such as viral hepatitis, drug-induced liver injury, or autoimmune hepatitis)*: 5% **Reasoning**: Acute liver failure is a rare but life-threatening condition characterized by rapid deterioration of liver function. The presence of jaundice, abdominal pain, nausea, vomiting, fatigue, and itching, along with the acute onset of symptoms, may indicate acute liver failure. However, the absence of altered mental status (hepatic encephalopathy) makes this less likely.

### E.6 GPT-4 THIRD ROUND

- *Hepatitis C (HCV)*: 35% **Reasoning**: Hepatitis C is indeed a major cause of the symptoms listed, including fatigue, abdominal pain, nausea, vomiting, jaundice, dark urine, and weight loss. Itching can also occur due to liver dysfunction.
- *Hepatitis B (HBV)*: 25% **Reasoning**: Hepatitis B shares many symptoms with Hepatitis C, such as fatigue, abdominal pain, nausea, vomiting, dark urine, jaundice, and itching. Testing for both HBV and HCV is warranted given the overlap in symptoms.
- *Cirrhosis*: 20% **Reasoning**: Cirrhosis can result from chronic hepatitis, alcohol abuse, or non-alcoholic fatty liver disease. The symptoms of fatigue, weight loss, abdominal pain, jaundice, dark urine, and itching strongly suggest advanced liver disease, which could be cirrhosis.
- *Obstructive Jaundice*: 15% **Reasoning**: Obstructive jaundice, caused by gallstones, pancreatic cancer, or bile duct obstruction, is consistent with jaundice, dark urine, itching, abdominal pain, nausea, vomiting, and weight loss. The absence of fever reduces the likelihood of an infectious cause like cholangitis.
- *Acute Liver Failure*: 5% **Reasoning**: Acute liver failure is less common but serious, characterized by rapid deterioration of liver function. The symptoms of jaundice, abdominal pain, nausea, vomiting, fatigue, and itching, along with the acute onset of symptoms, suggest this possibility. However, the absence of altered mental status (hepatic encephalopathy) makes it less likely.

### E.7 CLAUDE THIRD ROUND

- *Hepatitis C (HCV)*: 35% **Reasoning**: I slightly lowered the probability of HCV from 40% to 35%. While the combination of symptoms strongly suggests HCV, the absence of a few key risk factors (e.g., history of intravenous drug use or blood transfusions) in the given information makes me less confident in assigning a higher probability.

- *Hepatitis B (HBV)*: 30% **Reasoning**: I agree with the 30% probability assigned to HBV. The symptoms are indeed similar to HCV, and testing for both viruses is warranted.

- *Cirrhosis*: 20% **Reasoning**: I increased the probability of cirrhosis from 15% to 20%. The presence of jaundice, dark urine, itching, and weight loss, along with the other symptoms, strongly suggests advanced liver disease. Cirrhosis can be a consequence of chronic liver diseases like hepatitis B or C, which have been assigned higher probabilities.

- *Obstructive Jaundice*: 10% **Reasoning**: I agree with the 10% probability for obstructive jaundice. The absence of fever makes an infectious cause like cholangitis less likely, but the other symptoms still suggest a possible obstruction.

- *Acute Liver Failure*: 5% **Reasoning**: I agree with the 5% probability for acute liver failure. While the combination of symptoms could indicate acute liver failure, the absence of altered mental status (hepatic encephalopathy) makes this diagnosis less likely.

### E.8 JOINT FINAL ROUND

1. *Hepatitis C (HCV) - 35%* **Justification**: The combination of symptoms strongly suggests HCV, but without additional risk factors, a slightly lower probability is appropriate.

2. *Hepatitis B (HBV) - 30%* **Justification**: The symptoms are similar to HCV, and testing for both viruses is necessary.

3. *Cirrhosis - 20%* **Justification**: The presence of jaundice, dark urine, itching, weight loss, and other symptoms indicates advanced liver disease, making cirrhosis a more likely possibility.

4. *Obstructive Jaundice - 10%* **Justification**: The absence of fever reduces the likelihood of an infectious cause like cholangitis, but the other symptoms still suggest a potential biliary obstruction.

5. *Acute Liver Failure - 5%* **Justification**: While the combination of symptoms could indicate acute liver failure, the absence of altered mental status makes this diagnosis less probable.

### E.9 FINAL JOINT RECOMMENDATIONS TO GP

Gemini and GPT-4 jointly recommend the following additional symptom inquiries and lab tests to confirm a hepatitis diagnosis:

**Additional Symptom Inquiries:**

1. *Onset and Duration*: Determining the onset and duration of symptoms can clarify the illness timeline and its progression.

2. *Severity and Trend*: Evaluating symptom severity and whether they are worsening or improving aids in understanding the disease's trajectory and treatment responses.

3. *Associated Symptoms*: Checking for additional symptoms like nausea, vomiting, fever, joint pain, or urine color changes can pinpoint other hepatitis indicators and exclude other conditions.

**Recommended Lab Tests:**

1. *Liver Function Tests (LFTs)*: Critical for assessing liver health, LFTs evaluate enzyme levels such as alanine aminotransferase (ALT) and aspartate aminotransferase (AST), where abnormalities can signify liver inflammation.

2. *Hepatitis Panel*: This test checks for hepatitis A, B, and C viruses, vital for determining the specific type and guiding treatment strategies.

3. *Serology Testing*: Useful for distinguishing between acute and chronic hepatitis by identifying specific antibodies or antigens.

4. *Imaging Studies*: Ultrasound or MRI can provide visual insights into the liver's state, detecting inflammation, fibrosis, or cirrhosis, thereby complementing blood-based diagnostics.

This study demonstrates how EVINCE can identify potential misdiagnoses, elucidate the reasoning behind them, and suggest corrective actions. Traditionally, machine learning researchers rely on labeled data as "ground truth." However, research such as that by Newman-Toker et al. (2021) Newman-Toker et al. (2023a) from Johns Hopkins highlights that misdiagnosis is a pervasive issue in healthcare systems worldwide. These erroneous diagnoses, often assumed to be ground truth, can be perpetuated by supervised learning algorithms, thereby compounding the problem. EVINCE's dialogue capabilities provide insights into the decision-making process and highlight missing information, helping to rectify erroneous predictions and redefine the ground truth.

## APPENDIX F: EXPLAINABILITY AND RECTIFICATION

The power of EVINCE lies not only in its enhanced accuracy but also in its ability to elucidate the decision-making process and identify missing information, providing critical insights to correct errors.

Although the final joint prediction for Hepatitis C reached a high consensus of 37.5%, it deviates from the actual condition of Jaundice, which the Kaggle dataset reports with 10% confidence. EVINCE provides general practitioners with alerts and suggests remedial actions (see Appendices D.8 and E.9) to address this discrepancy. Recommended actions include querying additional symptoms from the patient and conducting specific laboratory tests.

EVINCE initiates debates with high contentiousness, encouraging dual prediction entropy between LLMs, as supported by the EDT theorem. It utilizes normalized mutual information (MI) to track shared knowledge accumulation throughout the debate, while Wasserstein distance (WD) and Jensen-Shannon divergence (JSD) quantify dissimilarity between LLM predictions.

These metrics (EDT, WD, JSD, MI) provide a comprehensive view of debate progress. WD and JSD assess the potential for further communication and refinement, while MI monitors shared understanding, aiding in determining the optimal stopping point.

The asymmetric nature of KL divergence and cross entropy warrants further investigation. Despite eventual convergence in our case studies, discrepancies observed in the second round (one direction increasing while the other decreases) suggest potential value in exploring asymmetric information. Future work will re-evaluate the use of these metrics if asymmetry proves beneficial.

Besides generating final joint disease predictions, EVINCE provides:

- Recommendations for additional symptom inquiries and lab tests to improve accuracy.

- Suggestions to query symptom onset, duration, severity, trends, and associated symptoms (documented in Appendices D.8 and E.9).

These recommendations have been verified by general practitioners to be valuable.

## APPENDIX G: COMPLEMENTARY EXPERIMENT — DEBIASING NEWS ARTICLES

This experimental framework aims to assess the feasibility of both detecting biases in textual content and implementing effective mitigation strategies. The first experiment focuses on bias detection, while the second explores the generation of balanced textual outputs as a corrective measure, moving beyond the limitations of prior studies that primarily focused only on identification.

| News # | Categories | Negative | W. Negative | Neutral | W. + | Biases (DR,DS,SR) | Source |
|---|---|---|---|---|---|---|---|
| D1* | Civil Rights | - | D,R,S,c | g | - | 0,0,0 | HuffPost |
| D2* | Civil Rights | D,S | - | R,c,g | - | 2,0,2 | HuffPost |
| D8 | Civil Rights | D | - | S,c,g | R | 3,2,1 | BBC |
| D31 | Environment | D | - | R,S,c,g | - | 2,2,0 | CNN |
| D37 | Politics | - | D,R,S,c,g | - | - | 0,0,0 | Yahoo |
| D69 | Healthcare | D,c | g | R,S | - | 2,2,0 | Breitbart |
| D81* | Economy | - | D,S | R,c | g | 1,0,1 | Breitbart |
| D98 | Economy | D,S,c,g | R | - | - | 1,0,1 | Breitbart |
| D101 | Education | c | D.S | R,g | - | 1,0,1 | New York Times |
| D106 | Election | - | g | D,R,S,c | - | 0,0,0 | USA Today |
| D109 | Elections | - | D,S,c,g | R | - | 1,0,1 | Reuters |
| D157 | International | - | D,S,c | R,g | - | 1,0,1 | New York Times |
| D174 | International | - | S,c | D,R,g | - | 0,1,1 | LA Times |
| D188 | National Security | - | S,c,g | D,R | - | 0,1,1 | Wall Street Journal |
| D278 | Civil Rights | - | D,S,c | R,g | - | 1,0,1 | Fox News |
| D336 | Politics | - | - | D,R,S,c,g | - | 0,0,0 | New York Times |
| Total | | | | | | 15,8,11 | |

Table 3: Comparison of bias assessments among Democrats (D), Republicans (R), and EVINCE (S), plus Claude (c) and GPT-4 baselines (g). It is observed that R and S are frequently placed to the right or in alignment with D, and only on two occasions does D precede S (highlighted in red). The ratings of the GPT-4 baseline (g) and EVINCE (S) exhibit an average gap of 0.6875, highlighting the substantial debiasing effectiveness of EVINCE.

To establish a baseline, we used Claude and GPT-4 to generate initial results. For experimenting with EVINCE, we used two instances of GPT-4, as Claude appeared prone to easily shifting its predictions (discussed shortly).

## G1: EXPERIMENT #1: BIAS DETECTION

The aim of this experiment is to evaluate if personal ideology may affect annotations, and can EVINCE help flag and rectify the biases.

**Dataset** This study utilizes a unique dataset of 619 news articles (54.3% about Democrat scandals, 45.7% about Republican scandals) selected from a larger 2013 repository of 14,033 articles compiled by fifteen reputable news organizations Budak et al. (2016). These articles span diverse topics including civil rights, healthcare, elections, and national security, offering a comprehensive view of political coverage. Please visit Anonymous (2024) for links to the full set of news articles.

**Value of Partisan Annotations** The dataset's distinctive feature is its ground-truth labels provided by annotators with declared political affiliations. Through Amazon Mechanical Turk, 749 qualified U.S. workers, each annotating up to 1,000 randomly selected articles, classified articles on a five-point scale from 'negatively

biased' to 'positively biased' Budak et al. (2016). Crucially, each scandal article in our subset received independent classifications from both Democrat and Republican annotators.

**Sufficiency of Current Annotations**   The current annotator pool provides a robust foundation for bias analysis for several reasons:

Natural Partisan Division: The dataset uniquely captures genuine political biases through annotators who self-identify as Democrats or Republicans, offering authentic opposing viewpoints that would be difficult to replicate artificially. Balanced Coverage: Each article receives evaluations from both political perspectives, creating natural "disagreement pairs" that reveal how political affiliation influences content interpretation.

Qualified Annotators: The original study employed rigorous qualification criteria for annotators, ensuring high-quality, considered judgments rather than casual opinions. Scale and Diversity: With 749 annotators across the full dataset, the annotations represent a broad spectrum of political viewpoints within each party, capturing intra-party variations in addition to inter-party differences.

This dataset's partisan annotations serve as an ideal testbed for our study, as they allow us to compare LLM-generated perspectives with human partisan viewpoints Evaluate EVINCE's ability to bridge opposing political interpretations Assess bias detection and mitigation strategies against clear partisan baselines

The original study Budak et al. (2016) revealed significant patterns in partisan perception: Republican annotators often perceived news about Republican scandals as negatively biased, while Democrat annotators viewed such coverage as neutral, indicating satisfaction with its perceived fairness. These documented patterns provide a valuable benchmark for evaluating EVINCE's bias detection capabilities. Adding more annotators would not necessarily enhance the dataset's utility, as the current partisan division already captures the fundamental dynamics of political bias in news interpretation. Instead, our focus is on leveraging these existing high-quality annotations to demonstrate how EVINCE can identify, understand, and help mitigate these well-documented partisan biases.

### G1.1 RESULTS ON DEMOCRAT SCANDALS

We apply EVINCE to analyze 619 news articles, comparing its labels with the dataset's provided ground truth. Additionally, we compare the results from EVINCE with the baseline generated through prompting Claude and GPT-4.

Table 3 compares the judgments of EVINCE (S), Republicans (R), and Democrats (D) on 16 representative articles (spanning different news sources and subjects) concerning "Democrat Scandals." The one-shot ratings from Claude are marked with lowercase 'c,' while those from GPT-4 are marked with lowercase 'g.'

Claude's judgments were found to be inconsistent, with identical prompts producing varying ratings, leading us to exclude further discussion of its outcomes. In contrast, GPT-4's one-shot ratings are stable but occasionally diverge from EVINCE. In 3 out of 16 articles, the rating difference exceeds one scale point. In these cases (D1, D2, and D81), EVINCE initiated further dialogue and successfully persuaded GPT-4 to revise its ratings. A complete debate on D1 is provided in Appendix F, illustrating how EVINCE modulates contentiousness and tracks the progression of metrics across rounds. Table 3 shows that after dialogue, EVINCE gains over the baseline performance of GPT-4 by 11 out of 16, or 0.6875 scale. This improvement is substantial. as the gap between R and D annotators is one scale (shown in Figure 8).

As expected, Democrats' judgments are generally more negative than Republicans', with EVINCE's assessments typically falling in between, except for two cases. Notably, there's a 5-to-1 Democrat-to-Republican ratio in the "Negative" column and a 12-to-4 Republican-to-Democrat majority in the "Neutral" column.

Tables 7 and 8 in Appendix H provide detailed justifications for EVINCE's ratings. To further investigate bias, we examine two specific articles: one from HuffPost (rated far left by AllSides Bias Chart Allsides) and another from Breitbart (rated far right).

* *D8 — HuffPost (Left)*: EVINCE rates D8 (on the third row) as neutral, citing the article's direct presentation of facts and inclusion of diverse perspectives on NSA surveillance practices and global reactions. This contrasts with Democrat-leaning annotators, who view the article as negatively biased towards Democrats, while Republican-leaning annotators favor it for exposing a Democratic scandal.
* *D69 — Breitbart (Right)*: EVINCE assesses D69 as weakly negatively biased towards Democrats, emphasizing its neutral tone and broad range of perspectives on NSA surveillance. This diverges from Democrat-leaning annotators, who rate it as strongly negative, but aligns with Republican-leaning annotators who deem it neutral.

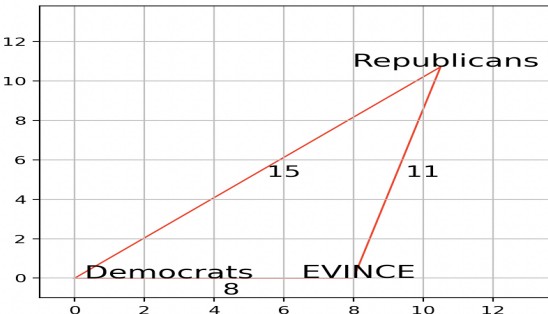

Figure 7: Distances Between D, R, and S.

In the last row of Table 3, we quantify the distances between annotations from Democrats (D), Republicans (R), and EVINCE (S), denoted as DR, DS, and SR respectively. Each unit of distance represents one step on the annotation scale (e.g., "Negative" to "Weak Negative"). Figure 7 visualizes these distances in a triangular plot. DR, the disparity between Democrat and Republican annotators, is the longest, followed by SR and then DS. This indicates EVINCE's statistical neutrality. These quantitative measures, along with the qualitative justifications in Appendix C, empower a human committee to decide whether adjustments or footnotes are warranted for polarized annotations.

### G1.2 RESULTS ON REPUBLICAN SCANDALS

Table 4 presents the bias assessments from EVINCE (S), Republicans (R), and Democrats (D) on articles related to "Republican Scandals." In contrast to the "Democrat Scandals" dataset, where Republican-leaning evaluations were more favorable, this dataset reveals a shift, with Republican-leaning assessments being notably more critical and Democrat-leaning assessments relatively neutral. The distance triangle for "Republican Scandals" mirrors the pattern seen in Figure 7, with the divergence between Republican and Democrat annotators being the largest (15). The distances between EVINCE and Democrat-leaning annotators (9) and between EVINCE and Republican-leaning annotators (11) are smaller, further highlighting EVINCE's relative neutrality.

Figure 8 illustrates the distribution of ratings for all scandals across four scenarios:

1) Democrat-leaning annotators rating Democrat scandals, 2) Republican-leaning annotators rating Democrat scandals, 3) Democrat-leaning annotators rating Republican scandals, and 4) Republican-leaning annotators rating Republican scandals.

The figure reveals a clear pattern: Democrat-leaning annotators tend to rate news about Democrat scandals more negatively, while Republican-leaning annotators exhibit similar negativity towards reports on Republican

| News # | Categories | Negative | W. Negative | Neutral | W. + | Biases (DR,DS,SR) | Source |
|--------|-----------|----------|-------------|---------|------|-------------------|--------|
| R1 | International | R,S | - | D | - | 2,2,0 | New York Times |
| R7 | National Security | - | - | D,R,S | - | 0,0,0 | New York Times |
| R15 | Economy | - | R | D,S | - | 1,0,1 | Huffington Post |
| R69 | Elections | - | D,S,R | - | - | 0,0,0 | Reuters |
| R124 | Gay Rights | R | S | D | - | 2,1,1 | Fox |
| R125 | Crime | - | R,S | D | - | 1,1,1 | Fox |
| R180 | Elections | - | - | D,R,S | - | 0,0,0 | AP |
| R191 | Elections | - | R | D,S | - | 1,0,1 | CNN |
| R214 | Gay Rights | R,S | - | D | - | 2,2,0 | Dailykos |
| R221 | Economy | - | R | D,S | - | 1,0,1 | Wall Street Journal |
| R233 | Economy | - | R,S | D | - | 1,1,0 | Fox |
| R235 | Civil Rights | D,R | - | S | - | 0,2,2 | Reuters |
| R269 | Healthcare | - | R | D,S | - | 1,0,1 | New York Times |
| R274 | Healthcare | - | R | D,S | - | 1,0,1 | USA Today |
| R280 | Politics | D,S | - | R | - | 2,0,2 | Fox |
| Total | | | | | | 15,9,11 | |

Table 4: Comparison of bias assessments. It is observed that D and S are frequently placed to the right or in alignment with R, and only on one occasion does D precede S (highlighted in red).

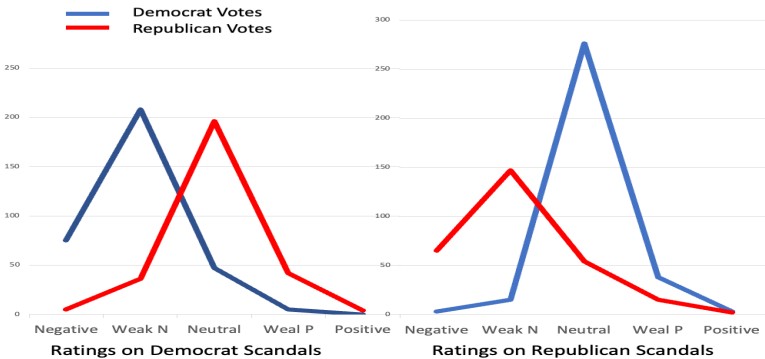

Figure 8: Bias Rating Distributions Show Strong Biases. D is more negative on how D scandals were reported (the sub-figure on the left), R is more negative on how R scandals were reported (the sub-figure on the right).

scandals. The gap between these ratings is approximately one class-label (e.g., between "weak negative" and "neutral"), highlighting a tendency within both parties to defend their own and criticize the opposition.

EVINCE, operating without emotional influence and refined through structured debate, consistently provides a more balanced, centrist perspective. This contributes to a more impartial discourse by mitigating partisan biases. EVINCE's justifications, documented in Appendix B, are transparent and reasonable. An editorial board can review these findings and decide whether to adjust labels or present both perspectives with explanations.

This experiment demonstrates that EVINCE effectively delivers centrist judgments supported by rationales. For a deeper understanding of EVINCE's bias assessment process, comprehensive justifications for each of the 31 analyzed articles are available in Appendix B.

## G.2 EXPERIMENT #2: BIAS MITIGATION

| Agent | Arguments |
|---|---|
| Agent A Round #1 | Christopher Columbus's voyages were pivotal in expanding European knowledge and trade. His journeys opened up the Americas to European colonization, significantly altering the course of history in four aspects: 1) expansion of knowledge and exploration, 2) positive economic impact, 3) historical significance, and 4) cultural exchange. |
| Agent B Round #1 | There are negative aspects and historical consequences of Columbus's actions. Key arguments are 1) indigenous suffering and genocide, 2) economic exploitation, 3) cultural erasure, 4) environmental impact, and 5) ethical considerations. |
| Agent A Round #3 | The severe consequences were a result of broader colonial policies and practices that followed. Evaluating historical figures requires understanding the context of their times. While modern ethical standards are important, they should not overshadow the historical significance and contributions of Columbus's voyages to global exploration and knowledge. |
| Agent A Round #4 | Agent A recognizes that mitigation efforts are part of an ongoing process. Historical scholarship is evolving, and the inclusion of indigenous perspectives, critical examinations of colonial legacies, and the acknowledgment of the complexities of historical figures like Columbus are essential steps toward more equitable and inclusive narratives. |

Table 5: Debate arguments leading to neutrality

This experiment illustrates EVINCE's ability to identify bias in text, provide reasoned justifications, and propose remediation through the integration of diverse perspectives. We demonstrate how EVINCE utilizes statistical and information theory metrics to facilitate multi-agent dialogue, circumventing the "maximum likelihood" trap inherent in next-token generation and uncovering information from multiple viewpoints.

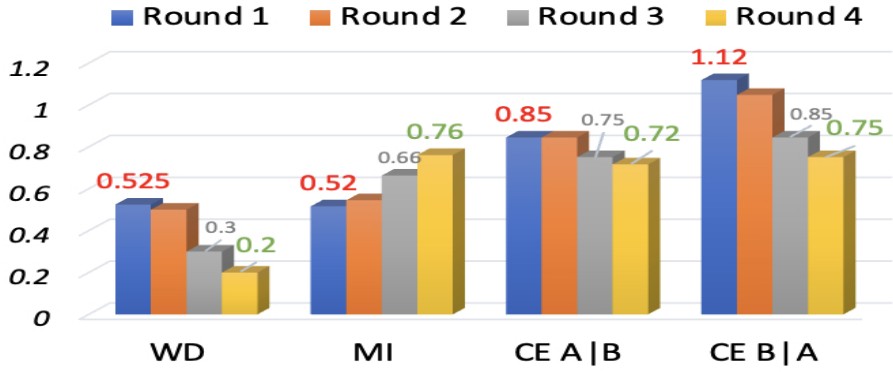

Figure 9: Convergence of all metrics, Wasserstein, normalized mutual information, normalized cross entropy

Using the example of the Euro-centric perspective on Christopher Columbus' Wikipedia page regarding his voyages to America, EVINCE employs two GPT-4 instances: Agent A, supporting the Euro-centric view, and Agent B, opposing it. Table 5 summarizes Agent A's key arguments and its evolving stance throughout the debate.

Guided by the maxims and entropy duality theorem from Section 2, we initiate the debate by prompting both agents to defend their positions rigorously and score each other's bias using a five-label distribution (negative, weak negative, neutral, weak positive, positive). Figure 9 tracks the dialogue's progress through Wasserstein distance (WD) Kantorovich (1942), normalized cross entropy (CE) Shannon (1948), and normalized mutual information (MI) Cover & Thomas (2006a). Initially, each agent is expected to perceive itself as neutral

and the other as biased. The debate concludes when the bias distributions converge and mutual information plateaus, indicating a shared understanding.

## G3 OBSERVATIONS AND EXTENDED FINDINGS

Our initial observation highlights a key challenge in working with LLMs: without explicit and repeated reminders of their assigned stance (pro-discovery or pro-encounter), GPT-4 instances can revert to default statistical behavior, evaluating their own arguments based on overall language patterns rather than the intended perspective. This was evident when Agent B, despite being assigned to support the Indigenous perspective, initially rated its own arguments as "positively biased." A reminder to adhere to its assigned role prompted a correction to "neutral," underscoring the importance of careful context management and reinforcement, especially given the limited token size of LLMs.

The second observation demonstrates a positive outcome of the debate process. The revised bias distributions, incorporating rational responses that acknowledge both positive and negative aspects of Columbus's voyages, show a shift towards a more balanced perspective. Agent A moves towards neutrality while acknowledging historical context, while Agent B maintains a critical stance but strives for balanced representation. This approach facilitates a deep and comprehensive understanding of Columbus's legacy.

## G4 SUMMARY OF EVINCE DEBATE ON NEWS D1

| # | Agent | Neg. D. | W. Neg. D. | N. | W. Neg. R. | Neg. R. | WD | KL | JS | Δ |
|---|-------|---------|------------|-----|------------|---------|------|-------|-------|-----|
| 1 | A | 5% | 15% | 50% | 25% | 5% | 0.45 | 0.316 | 0.081 | 90% |
|   | B | 10% | 10% | 25% | 35% | 20% | | | | |
| 2 | A | 7% | 13% | 40% | 30% | 10% | 0.47 | 0.226 | 0.056 | 70% |
|   | B | 5% | 10% | 20% | 40% | 25% | | | | |
| 3 | A | 5% | 10% | 35% | 35% | 15% | 0.10 | 0.016 | 0.004 | 30% |
|   | B | 5% | 10% | 30% | 35% | 20% | | | | |
| Fin | A | 5% | 10% | 30% | 35% | 20% | 0 | 0 | 0 | 10% |
|   | B | 5% | 10% | 30% | 35% | 20% | | | | |

Table 6: Debate Parameters between, A and B, two GPT-4 instances. Information metrics and WD all converge to zero in the final round. Contentiousness Δ decreasing as the metrics approach zero.

The news under debate is D1 listed in Anonymous (2024). Please refer to Table 6 for the confidence distributions of Agents A and B throughout the four-round debate (following Maxim #4). The metrics, Wasserstein Distance (WD), Kullback-Leibler Divergence (KL), and Jensen-Shannon Divergence (JS), consistently decrease, indicating convergence and leading to final agreement in the last round. Meanwhile, the level of contentiousness is modulated according to the metrics' progress, decreasing from high (90%) to medium, and eventually reaching a conciliatory level (30%) and then agreement.

## G5 APPROACH TO COMPUTING CONTENTIOUSNESS

We could define contentiousness as a function of the divergence metrics. Since KL, JS, and WD measure the difference or "disagreement" between two distributions, a larger divergence requires higher contentiousness level to bridge, while lower contentiousness corresponds to more agreement.

A simple linear mapping can convert these metrics into a normalized contentiousness score between 0 and 1. Here's a weighted formula to compute it:

$$\Delta = \alpha \cdot \frac{KL}{KL_{\max}} + \beta \cdot \frac{JS}{JS_{\max}} + \gamma \cdot \frac{WD}{WD_{\max}}, \text{where}$$

- $KL, JS, WD$ are the values of the divergence metrics for the round.
- $KL_{\max}, JS_{\max}, WD_{\max}$ are the maximum possible values for each metric (used for normalization).
- $\alpha, \beta, \gamma$ are weights that control the influence of each metric. For simplicity, we can set $\alpha = \beta = \gamma = \frac{1}{3}$ for equal influence.

We then scale the contentiousness to a percentage between 0% and 100%.

## G6 SUPPORTING ARGUMENTS

In the following, we document the supporting arguments made by the two agents in each round, illustrating how their positions evolved toward consensus.

### ROUND 1: INITIAL ASSESSMENTS

**Agent A:** Emphasized the article's attempt to maintain balance, with moderate negativity toward Republicans but largely neutral reporting. Recognized slight bias against Republicans in the framing of intra-party conflict.
**Agent B:** Contended that the article's structure and language choices leaned more negatively toward Republicans, emphasizing Issa's inconsistency and Republican discord. Proposed a less neutral distribution.

### ROUND 2: FIRST REBUTTAL AND COUNTER-REBUTTAL

**Agent A:** Adjusted the probability distribution, acknowledging the article's criticism of Republicans but maintained that the focus reflected evolving political dynamics rather than bias.
**Agent B:** Persisted that the narrative framed Issa more negatively, with selective quotes and framing shifting focus from the investigation to Republican disarray.

### ROUND 3: MODERATE TONE REASSESSMENT

**Agent A:** Moderated its position, accepting that subtle framing could influence perceptions. Reaffirmed that the reporting aimed for balance but acknowledged areas where Issa was framed less favorably.
**Agent B:** Adopted a more modest tone, agreeing that while the article was not overtly biased, subtle editorial choices did tilt the narrative slightly against Republicans.

### ROUND 4: FINAL AGREEMENT AND CONSENSUS

**Both Agents:** Reached consensus on the final distribution. Acknowledged that the article maintained surface-level neutrality, with subtle biases leading to a moderate negative lean against Republicans. Both agents converged on the same probability distribution for the final round.

## G7 JUSTIFICATIONS OF 31 BIASED ARTICLES

In Section 4, we note that EVINCE processed 31 news articles to assess their neutrality. In addition to the final decision, we detail the justifications EVINCE provides at the debate's end. These justifications are documented in four tables: Tables 7, 8, 9, and 10.

| # | Rating S,D,R | EVINCE Justifications |
|---|---|---|
| D1 | 4,4,4 | The article discusses the controversy surrounding the IRS's alleged targeting of conservative groups, focusing on Rep. Darrell Issa's stance against releasing full transcripts from IRS interviews. Issa argues that full disclosure would compromise the investigation, a position criticized by Rep. Elijah Cummings, who advocates for transparency. The debate highlights political tensions, with implications of bias in the handling of the investigation. The article portrays a complex interplay of political, ideological, and possibly historical biases, reflecting the contentious nature of the IRS scandal and its investigation. |
| D2 | 5,5,3 | The article discusses Michele Bachmann's statement that the IRS scandal, involving the targeting of conservative groups, undermines the credibility of the 2012 election. Bachmann accuses the Obama administration of lying and questions the impartiality of the IRS, particularly in relation to health care coverage for conservatives. The article reflects political and ideological biases, given its focus on Bachmann's perspective and the implications for the election's integrity and government trustworthiness. |
| | | The perception of bias in Michele Bachmann's statement about the IRS scandal primarily stems from the framing and implications of her comments. She directly links the IRS's actions to the credibility of the 2012 election and the Obama administration, suggesting a deliberate misuse of power for political gain. This perspective inherently carries ideological and political biases by casting the issue in a light that emphasizes misconduct by one political group over another, without presenting counterarguments or broader context that might mitigate or challenge that viewpoint. |
| D8 | 3,5,2 | The reporting appears to maintain a neutral tone by directly stating the facts and comments made by John Kerry and others involved without inserting opinionated language. It discusses the global reaction to the NSA's surveillance practices and includes Kerry's acknowledgment of overreach, as well as the international and domestic fallout from these revelations. The inclusion of various perspectives, including those from technology companies and international officials, alongside Kerry's comments, aims to provide a balanced view of the situation. |
| D31 | 3,5,3 | The CNN article appears to report on Ray Nagin's indictment with a focus on the factual and legal aspects of the case, without displaying overt biases related to politics, ideology, religion, gender, race, socioeconomic status, culture, geography, or history. By sticking closely to the details of the indictment and Nagin's public actions and statements, the article provides a comprehensive overview of the charges against him while maintaining journalistic neutrality. |
| D37 | 4,4,4 | The article outlines how Senate Democrats, led by Sen. Chuck Schumer, strategically navigated the border security issue to secure aid for Ukraine while potentially shifting the political narrative around immigration policy. Schumer's approach to integrate border security into the aid package discussions aimed to both address the issue and leverage political gain. It suggests a calculated maneuver to position Democrats favorably on border security and hold Republicans accountable for any failure to pass the legislation, demonstrating a nuanced political strategy in the face of complex legislative challenges. |
| D69 | 3,5,3 | The article has a clear perspective that favors religious liberty arguments against the HHS Mandate of Obamacare. It specifically highlights cases where the mandate was challenged on religious grounds, suggesting a bias towards those opposing the mandate. The framing and choice of sources, emphasizing victories against the mandate and quoting lawyers from organizations focused on religious freedom, contribute to a viewpoint that may not fully account for counterarguments or the broader context of healthcare policy. It leans towards a particular ideological stance, making it less of a neutral report. |
| D81 | 4,4,3 | The article's focus on the possibility of conservative-owned car dealerships being targeted for closures during the General Motors bailout could imply a certain bias by emphasizing a narrative of political victimization without presenting a comprehensive range of perspectives or evidence. It suggests a parallel with the IRS's targeting of Tea Party groups, which could lead readers to infer a broader pattern of political discrimination without definitive proof. The call for an investigation by the Congressmen is legitimate news, but the framing and selection of information could influence the reader's perception of the events. |
| D98 | 5,5,4 | Yes, the article itself exhibits bias by focusing solely on criticizing the media's treatment of Obama's vacations compared to Bush's, without offering a balanced view or acknowledging any reasons why coverage might differ. It selectively presents information to support its claim of a double standard, which is a characteristic of biased reporting. |
| D101 | 4,4,3 | The article describes President Obama's strategy to navigate through political controversies by focusing on legislative actions and executive orders that bypass Republican opposition. It highlights the White House's efforts to concentrate on immigration reform, budget deals, healthcare law implementation, and keeping student loan rates low. The narrative suggests a proactive approach to governance amidst challenges, aiming to draw a contrast with what is portrayed as Republican political gamesmanship. This portrayal might be viewed as leaning towards a positive depiction of Obama's administration's efforts to prioritize policy over politics. It presents his efforts in a positive light, emphasizing a proactive and policy-driven approach amidst challenges. |

Table 7: The First 9 of 16 Democrat Scandals News Articles Rated by EVINCE and its Justifications. The rating column starts with EVINCE's rating, the Democrat rater (in blue), and then Republican rater (in red).

| D106 | 3,3,3 | The article reports on former Detroit Mayor Kwame Kilpatrick's sentencing to 28 years in prison for public corruption, emphasizing the gravity of his crimes against the city's welfare. It contrasts Kilpatrick's actions with the impact on Detroit, highlighting the judicial and public response to his extensive criminal activities. The reporting focuses on factual recounting of the trial's outcome, Kilpatrick's and his co-defendant's crimes, and the broader implications for Detroit, without evident bias towards political, ideological, or other specific perspectives. |
|---|---|---|
| D109 | 4,4,3 | The article's bias primarily stems from its focus on internal Democratic opposition to Lawrence Summers' Federal Reserve Chair nomination, highlighting a lack of unity and strategy within the party and the White House's mismanagement of the nomination process. It suggests an underestimation of the opposition's seriousness by the White House, portraying the administration in a somewhat negative light for not engaging more proactively with concerned Senate Democrats. |
| D157 | 4,4,3 | The article discusses the challenges in U.S.-Germany intelligence relations following revelations of U.S. surveillance on Chancellor Merkel. Despite efforts to rebuild trust, significant differences in surveillance philosophies persist, with the U.S. prioritizing security interests and Germany emphasizing privacy and alliance values. The situation reflects broader tensions in U.S. relations with allies over privacy and surveillance practices.
The article's framing might suggest a bias towards highlighting the challenges and frictions in the U.S.-Germany intelligence relations, particularly emphasizing Germany's privacy concerns and skepticism towards U.S. surveillance practices. It portrays the U.S. stance as unyielding and contrasts this with Germany's emphasis on privacy and legal constraints, potentially casting the U.S. in a more negative light regarding international surveillance and cooperation. |
| D174 | 4,3,3 | The article reports on House Speaker John Boehner and House Majority Leader Eric Cantor, both Republicans, expressing support for President Obama's proposal to authorize military action against Syria in response to the use of chemical weapons. This bipartisan backing is seen as crucial for Obama in gaining Congressional approval. The leaders emphasized the need for the U.S. to stand against such behavior internationally and the importance of the administration convincing both Congress and the American public of the strike's necessity.
The reporting appears balanced, focusing on factual statements and actions by political leaders regarding support for military action in Syria. It provides viewpoints from both Republican and Democratic leaders, their reasoning, and the challenges involved in convincing Congress and the American public. The emphasis on bipartisan support and the detailed reporting of various opinions and statements help maintain a neutral tone without apparent bias towards one political viewpoint or another. |
| D188 | 4,3,3 | The article reports that Hillary Clinton received warnings about security threats in Benghazi before the 2012 attack through emails. These were part of around 300 emails released by the State Department, which also show Clinton's responses and thoughts during the aftermath. The political controversy regarding the Obama administration's initial assessment of the attack's cause is also mentioned, with references to Susan Rice's statements on talk shows.
The reporting may exhibit bias through selective emphasis on Clinton's receipt of warnings about security threats in Benghazi and her responses, potentially framing her actions or inactions in a negative light. The focus on the political controversy and the administration's initial assessment of the attack's cause could also influence perceptions of responsibility or competence. |
| D278 | 4,4,3 | The article discusses a House committee investigation into whether Attorney General Eric Holder misled Congress regarding the Department of Justice's pursuit of journalists' personal records. It highlights Holder's previous testimony claiming no involvement in prosecuting press for material disclosure, contrasted with the DOJ's actions against Fox News reporter James Rosen, who was labeled a potential "co-conspirator" but not charged.... The article's tone suggests potential bias by emphasizing allegations that Eric Holder may have misled Congress, primarily focusing on criticisms from Republican figures and conservative critics. It uses specific incidents to question Holder's integrity and the Justice Department's actions under his leadership, possibly framing the narrative to highlight governmental overreach and lack of transparency. |
| D336 | 3,3,3 | The article reports on a scandal involving overcharging the Navy for ship supplies by Inchcape Shipping Services, owned by the government of Dubai, expanding beyond the Pacific Fleet to include Navy operations in the Middle East, Africa, and South America. The Justice Department is investigating allegations of Inchcape, through subcontractors, overcharging millions of dollars. This follows a separate criminal investigation into Glenn Defense Marine Asia for bribing Navy officials for overbilling. The Navy has suspended Inchcape from new contracts, marking a significant issue of fraud and corruption within Navy contracting. The reporting appears balanced, focusing on the factual aspects of the scandal and the investigation by the Justice Department. It cites sources and includes statements from involved parties, aiming to present a comprehensive view of the situation without apparent bias toward any side. |

Table 8: The Last 7 of 16 Democrat Scandals News Articles Rated by EVINCE and its Justifications. The rating column starts with EVINCE's rating, the Democrat rater (in blue), and then Republican rater (in red).

| # | Rating S,D,R | EVINCE Justifications |
|---|---|---|
| R1 | 5,3,5 | The editorial from The New York Times Editorial Board exhibits a clear bias against the Republican party's handling of the Benghazi attack investigation, framing it as politically motivated rather than a genuine effort to address security failures or seek accountability. The language and tone used in the editorial are critical and dismissive of the Republicans' actions, suggesting a political and ideological bias. While editorials are inherently opinion-based and are expected to take a stance, this piece clearly communicates a stance that is critical of the Republicans' focus on Benghazi, suggesting a lack of neutrality in its assessment of the motives and actions surrounding the investigation. |
| R7 | 3,3,3 | The article reports on allegations by Senator Mitch McConnell that his campaign headquarters were wiretapped, with the FBI investigating these claims. A recording of McConnell's team discussing potential attacks on Ashley Judd, who was considering running against him, was released by Mother Jones. McConnell accused the political left of this action, describing it as a "Nixonian move." The recording included discussions on various strategies to undermine potential opponents, highlighting a focus on Judd's personal struggles and political views. The controversy has prompted responses from both Republican and Democratic officials, reflecting the tense political atmosphere. |
| R15 | 3,3,4 | The report appears to present the information neutrally, stating both President Obama's rejection of the Republican proposal and the subsequent pushback from Republican sources who claim otherwise. It includes statements from both sides and provides context about the ongoing negotiations without overtly favoring one perspective over the other. Therefore, based on the information provided, the report does not appear to exhibit bias. |
| R69 | 4,4,4 | The report discusses how young Republicans are seeking a different message for elections, emphasizing a departure from divisive social issues and a focus on fiscal responsibility, national defense, and energy advancement. Selection Bias: The article primarily focuses on young Republicans who are seeking a different message for the party. It doesn't provide as much insight into young Republicans who may still align with traditional conservative values, which could create a slight bias toward the viewpoints of those seeking change. Language Bias: Certain language choices, such as describing divisive social issues as "anti-abortion, anti-gay, and anti-environment stances," may reflect a bias toward more progressive viewpoints on these issues. A more neutral description might be "positions on abortion, same-sex marriage, and environmental policy." Source Bias: The perspectives provided in the article are mainly from young Republicans themselves. While including these voices is essential, the article could benefit from additional perspectives from political analysts or experts to provide more context and balance. |
| R124 | 4,3,5 | The article provides a factual recount of the events surrounding Dr. Ben Carson's comments on gay marriage and the backlash from Johns Hopkins students. It maintains a relatively neutral tone and allows for the inclusion of multiple perspectives, including Carson's own response and apology. However, the lack of in-depth analysis into the implications of Carson's comparisons or the broader context of the gay marriage debate might leave readers without a complete understanding of the controversy's depth. Furthermore, the article does not explicitly offer viewpoints opposing Carson's beyond the students' petition, which could be seen as a form of omission bias. Yet, it does not overtly favor Carson or dismiss the students' concerns, striving instead to report on the unfolding situation. |
| R125 | 4,3,4 | The news article on the Zimmerman verdict and the subsequent calls for a Florida boycott provides a complex mix of reporting that touches on several sensitive themes, including political and ideological biases, as well as racial and socio-economic considerations. While the article attempts to cover a contentious and complex issue by incorporating diverse viewpoints, the focus on political figures, racial dynamics, and economic repercussions could introduce biases in how the information is perceived. To mitigate these biases, the reporting could benefit from a more in-depth exploration of the legal and historical contexts, a broader range of perspectives, and a careful consideration of how the information presented might influence public perception. |
| R180 | 3,3,3 | The AP article provides a balanced account of the corruption scandal involving three politically involved sisters in Pennsylvania. It maintains an objective tone while delivering comprehensive background information that situates the legal outcomes within the broader context of the Orie family's public and political life. The reporting is factual and avoids taking sides, instead focusing on the legal facts and the personal and political fallout for the individuals involved. |
| R191 | 3,3,4 | The CNN article attempts to navigate the complexities of Mark Sanford's political comeback attempt in a balanced manner, focusing on the potential impact of his past infidelity on female voters and the broader electoral implications. By presenting viewpoints from both sides of the political aisle, incorporating a range of sources, and providing necessary background and context, the article strives for neutrality. While the subject matter inherently involves sensitive and potentially polarizing topics, the article's approach is to inform readers about the electoral dynamics at play without overt bias. |

Table 9: The First 8 of 15 Republican Scandals News Articles Rated by EVINCE and its Justifications. The rating column starts with EVINCE's rating, the Democrat rater (in blue), and then Republican rater (in red).

| # | Rating S,D,R | EVINCE Justifications |
|---|---|---|
| R214 | 5,3,5 | The Daily Kos article clearly exhibits political and ideological bias, with a tone and language that are critical of the Republicans. It uses the incident involving Dave Agema to argue that the party's efforts to rebrand itself as more tolerant and inclusive are undermined by its members' actions. While the article includes factual information regarding the incident and the party's response, its presentation and commentary are aligned with a progressive viewpoint, aiming to highlight and criticize perceived contradictions and failures within the Republican Party. This approach is consistent with opinion journalism but introduces bias through its critical tone, selective presentation of information, and framing of the incident as emblematic of broader issues within the party. |
| R221 | 3,3,4 | "Hurricane Christie" presents Governor Chris Christie's critique of House Republicans in a manner that emphasizes party conflict and personal betrayal. The dramatic framing, choice of language, and focus on internal discord may introduce bias by portraying Christie's actions in a specific light and emphasizing the divide within the Republican Party. The article's approach to presenting these events can influence readers' perceptions, potentially leading them to see the situation through a lens of heightened drama and internal strife. |
| R233 | 4,3,4 | While the article attempts to cover the last-ditch efforts by House Republicans to avert a government shutdown and the standoff with Senate Democrats, the framing and language used may introduce a bias towards portraying the Republican efforts in a more favorable light. By emphasizing the Republican narrative of seeking negotiation and characterizing the Democratic response as dismissive, the article could be perceived as leaning towards a particular political perspective. The inclusion of quotes and perspectives from both sides does provide a degree of balance, but the overall presentation and emphasis could influence readers' perceptions of the shutdown negotiations. |
| R235 | 3,5,5 | Without knowledge of the author or publication, this text attempts to navigate a complex and sensitive story by providing details from multiple sources, including the main figures involved, political watchdog groups, and law enforcement. It balances the serious allegations with responses from the accused, background information, and the current status of investigations. While the focus on unsubstantiated claims could inherently sway public opinion, the article's inclusion of diverse perspectives and context aims to mitigate overt bias. |
| R269 | 3,3,4 | The article reports on President Obama's efforts to address the government shutdown, his challenge to Speaker John Boehner regarding the passage of a budget measure, and the broader context of the political standoff over the Affordable Care Act and the debt ceiling. To evaluate the article for bias, we'll examine it against various criteria... |
| R274 | 3,3,4 | The article presents a relatively balanced view of the internal GOP conflict over the strategy to defund the ACA, highlighting arguments from both sides of the debate within the party. It focuses on the political and strategic dimensions of the issue, providing insights into the perspectives of key figures and factions within the Republican Party. While the article could potentially be seen as emphasizing party divisions, which might align with certain political narratives, it does so in the context of exploring a significant and newsworthy internal debate. The absence of discussion on the socioeconomic, cultural, and historical contexts of the ACA debate, however, suggests areas where the reporting could be expanded to provide a more comprehensive view of the issue. The article strives to present a comprehensive view of the government shutdown, the debate over the Affordable Care Act, and the looming debt ceiling crisis by including perspectives from both the Obama administration and Republican leaders. While there is an emphasis on Obama's attempts to resolve the situation and his calls for Congress to act, the inclusion of Republican viewpoints and the mention of the piecemeal funding bills passed by the House attempt to provide a balanced perspective. The reporting appears to aim for neutrality by focusing on the facts of the political standoff and the implications for federal operations and the nation's financial credibility. |
| R280 | 5,5,3 | The article from Fox News by Jay Sekulow, titled "Obama's fingerprints all over IRS Tea Party scandal," presents a viewpoint that directly implicates President Obama in the IRS scandal involving the targeting of conservative groups. The author argues that the scandal was not only known but encouraged by senior IRS officials, Congressional Democrats, the White House, and further fueled by the mainstream media. To assess the article for bias, let's evaluate it against various criteria: The article "Obama's fingerprints all over IRS Tea Party scandal" demonstrates clear political and ideological biases, with a narrative constructed to directly implicate President Obama in the IRS targeting scandal. By selectively quoting Obama and drawing connections to actions by the IRS, the article aims to present a cohesive narrative that places responsibility for the scandal on the president. This framing serves to reinforce the viewpoint of those who see the actions as politically motivated and indicative of broader issues of governance and accountability under the Obama administration. The choice of language, historical comparisons, and the leveraging of the author's and platform's ideological stances contribute to a biased presentation of the events surrounding the IRS scandal. |

Table 10: The Last 7 of 15 Republican Scandals News Articles Rated by EVINCE and its Justifications. The rating column starts with EVINCE's rating, the Democrat rater (in blue), and then Republican rater (in red).

