# OpenReview forum: "EVINCE: Optimizing Adversarial LLM Dialogues via Conditional Statistics and Information Theory"
_ICLR.cc/2025/Conference — Submitted to ICLR 2025_

### Official Review · Reviewer_xMVe · 2024-10-30

**Soundness:** 3
**Presentation:** 2
**Contribution:** 2
**Rating:** 5
**Confidence:** 3

**Summary:**

This paper proposes EVINCE, a dialogue framework for LLMs that leverages adversarial debate, information theory, and conditional statistics to enhance model performance in prediction accuracy, robustness, and adaptability. The framework introduces the concepts of Inclusiveness Exploration, Information Flow Dynamics, and Reasoning Quality and Coherence, providing a structured debate mechanism that balances diverse exploration with convergence. The work emphasizes the framework’s potential for AGI development by addressing common LLM limitations, such as hallucination and bias, through its structured interaction methodology.

**Strengths:**

The integration of conditional statistics and information theory into LLM adversarial dialogues is novel, especially with the dual entropy framework balancing exploration and prior adherence. The empirical results validate EVINCE’s improvements in diagnostic accuracy and reasoning quality.

**Weaknesses:**

1. Instead of disease diagnosis, expanding tests across more diverse domains would strengthen the claim of EVINCE’s general applicability.

2. The quality evaluation of arguments is important for the EVINCE framework and dependent on the proposed CRIT scores, so more insight into how CRIT scores are generated and their potential variability across domains could improve reliability.

3. Wasserstein distance and mutual information could be computationally intensive. How about comparing them to other similar but more efficient metrics and adding more discussion about this.

**Questions:**

1. What is the impact of the predictions on varying contentiousness and entropy values? Are there any recommended default settings for these in specific tasks, or is parameter tuning always necessary?

2. Are there any findings of differences in performance between different LLM backbones?

---

> ### Author Response · Authors · 2024-11-20
>
> Dear Reviewer,
>
> Thank you for your thoughtful and constructive feedback. We appreciate your insights and the opportunity to address the specific points you raised.
>
> A. Responses to Weaknesses:
>
> 1. Empirical Studies and Domain Applications:
> Due to the anonymous review policy, we have refrained from citing all our previous works, as doing so would make it obvious who we are. Additionally, the page limitation has constrained our ability to provide comprehensive details about all the domains we have explored. However, we want to emphasize that we have conducted empirical studies across multiple domains beyond healthcare, including ethics alignment (accepted by NeurIPS 2024), debiasing news articles, geopolitical analysis, investment, and sales planning. These applications showcase the adaptability and effectiveness of EVINCE in a wide range of contexts. If the reviewer would like, we can include an appendix in the November 27 revision summarizing these works and their results.
>
> 2. CRIT Documentation:
> Detailed documentation of CRIT is provided in the appendix of our submission. Additionally, the full paper on CRIT, which outlines its methodology and applications, is cited in the references.
>
> 3. Computational Efficiency:
> In the broader picture, the computational cost of EVINCE is negligible, representing less than one in a million in terms of overall system efficiency. Specifically, EVINCE typically involves fewer than five predictions per outcome. Comparing two distributions with, for instance, 5 + 5 possible outcomes results in a quadratic complexity, but this cost is minimal because N is small, and is negligible relative to the overall process.
>
> B. Responses to Questions:
>
> 1. Impact of Predictions on Contentiousness and Entropy Values:
> Contentiousness and entropy are critical in balancing exploration and convergence during dialogue. Based on our experiments, we observed the following patterns:
>
> - High contentiousness (e.g., 0.7–0.9): Effective in the early stages of dialogue to encourage diverse exploration and uncover inconsistencies in the LLMs’ reasoning. This phase generates new insights by probing opposing viewpoints rigorously.
> - Moderate contentiousness (e.g., 0.4–0.6): Facilitates the transition from exploration to refinement, resolving contradictions and consolidating arguments.
> - Low contentiousness (e.g., 0.2–0.3): Supports consensus-building and final decision-making, ensuring productive outcomes.
>
> Our experiments across five domains have demonstrated that these three levels of contentiousness are sufficient to achieve strong results. Remarkably, LLMs align their linguistic behaviors dynamically based on these values, showcasing their adaptability and self-regulation during dialogue.
>
> 2. Differences in Performance Across LLM Backbones:
> In our experiments, EVINCE was tested using GPT-4, Claude, Gemini, and combinations of these models. The following observations were made:
>
> - Comparable Quality LLMs (e.g., GPT-4 vs. Claude): Result in balanced performance, with no significant dominance in reasoning or argument quality. These setups facilitate productive debates with evenly matched contributions.
> - Differing Quality LLMs (e.g., GPT-4 vs. smaller models): Skew dialogue dynamics, with the stronger model often dominating the discourse. This imbalance highlights the importance of using agents with comparable capabilities in adversarial setups to ensure fair and meaningful debates.
>
> We will include these findings in the revised manuscript, along with an expanded discussion on how different LLM backbones influence dialogue dynamics and outcomes.

---

> > ### Comment · Reviewer_xMVe · 2024-11-26
> >
> > Thanks for the rebuttal from authors. However, some of my concerns still exist. More intuitive experimental results about the raised questions are expected. It is regrettable that there are no analysis results in the rebuttal. Thus, I decide to keep the current rating.

---

> ### Author Response · Authors · 2024-11-26
>
> Dear Reviewer,
>
> I believe we have responded to your comments with substantial revisions to the paper. Specifically, we added an 11-page experiment on debiasing news articles to provide further evidence. We also clarified how CRIT is used to normalize prediction rankings.
>
> The CRIT work was published in the open domain in March 2023, and Appendix B provides a good summary.
>
> 𝗪𝗼𝘂𝗹𝗱 𝘆𝗼𝘂 𝗽𝗹𝗲𝗮𝘀𝗲 𝗿𝗲𝘃𝗶𝗲𝘄 𝘁𝗵𝗲 𝗿𝗲𝘃𝗶𝘀𝗲𝗱 𝗽𝗮𝗽𝗲𝗿? 𝗜 𝗮𝗽𝗼𝗹𝗼𝗴𝗶𝘇𝗲 𝗳𝗼𝗿 𝗻𝗼𝘁 𝗻𝗼𝘁𝗶𝗳𝘆𝗶𝗻𝗴 𝘆𝗼𝘂 𝗲𝗮𝗿𝗹𝗶𝗲𝗿 𝗮𝗯𝗼𝘂𝘁 𝘁𝗵𝗲 𝗻𝗲𝘄 𝘃𝗲𝗿𝘀𝗶𝗼𝗻, 𝘄𝗵𝗶𝗰𝗵 𝗜𝗖𝗟𝗥 𝗽𝗲𝗿𝗺𝗶𝘁𝘀 𝘀𝘂𝗯𝗺𝗶𝘀𝘀𝗶𝗼𝗻 𝗼𝗳 𝘂𝗻𝘁𝗶𝗹 𝟭𝟭/𝟮𝟳.
>
> The following addresses your concerns in four respects.  (𝙋𝙚𝙖𝙨𝙚 𝙡𝙚𝙩 𝙪𝙨 𝙠𝙣𝙤𝙬 𝙬𝙝𝙖𝙩'𝙨 𝙢𝙞𝙨𝙨𝙞𝙣𝙜?)
>
> %%
> A. Case Studies:
> We have refined Section 3 and added additional case studies in Appendix G, including the following:
>
> - A step-by-step dialogue example showing how EVINCE modulates contentiousness to balance exploration and exploitation
> - Analysis of how CRIT evaluates and calibrates reasoning quality
> - Examples where LLM collaboration identified potential misdiagnoses in the ground truth
> - Concrete instances of news debiasing using EVINCE (Appendix G)
>
> %%
> B. LLM Quality Requirement (already in Maxim 1):
> We state clearly in Dual Entropy Theory (DET), emphasizing that two equally competent LLMs, at least at the level of GPT-4, are required to achieve good results.
>
> The better the LLM backbone, the higher the quality of all NLU performance is expected. According to Demis Hassabis (DeepMind CEO), end-to-end solutions are preferable to ad hoc or human heuristics for addressing NLU tasks, as small-data approaches are not competitive. The reason for having two equally competent LLM instances is to avoid an asymmetric situation where one LLM dominates the other. CRIT is effective in this framework to evaluate reasoning quality via the Socratic methods.
>
> %%
> C. LLM initialization (already in Maxim 1 and 2):
>
> Stated in the paper, the initial condition is always set to be 1) one high in confidence distribution entropy and the other low, 2) one in high temperature (exploring) and the other low in temperature (exploiting), and 3) start from contentiousness = 90%. Subsequently, contentiousness will be modulated based on the quality metrics employed. The key parameter of the three is contentiousness. It was explained very thoroughly and also in the cited SocraSynth paper. Using information-theory metrics to automatically adjust the contentiousness level from combative to conciliatory dialogue is one of the novelties of this work. Traditional MAD methods often generate non-productive and repetitive arguments, due to the lacking ability to modulate the linguistic behavior of the participating LLMs (stated in the intro of the paper).  If two people in a debate maintain a consistently high, combative linguistic behavior throughout, the debate may devolve into an unproductive quarrel.
>
> Again, the following (from C.1 to C.3) summarizes EVINCE's modulation insights:
>
> C.1. Dynamic Contentiousness Modulation:
>
> EVINCE does not simply place two LLMs in a high-intensity, combative debate throughout.
> Instead, it uses metrics like entropy, mutual information, and divergence to continuously assess the quality and progress of the dialogue.
> Based on these metrics, EVINCE can automatically adjust the "contentiousness" level of the LLMs' linguistic behavior.
> It can start with a high contentiousness setting (90% in the initial condition) to encourage exploration and exchange of diverse perspectives.
> But as the dialogue progresses and the LLMs converge, EVINCE will gradually reduce the contentiousness to promote a more cooperative, conciliatory tone.
> This dynamic modulation prevents the debate from deteriorating into an unproductive quarrel, as you rightly pointed out.
>
> C.2. Balancing Exploration and Exploitation:
>
> The contentiousness modulation is a key innovation that allows EVINCE to strike a balance between exploration (diverse perspectives) and exploitation (convergence on the best solution).
> High contentiousness early on enables the LLMs to thoroughly canvass the problem space and consider multiple hypotheses.
> As the dialogue becomes more focused, lower contentiousness encourages the LLMs to cooperatively refine and consolidate their findings.
>
> C.3. Fostering Productive, High-Quality Debates:
>
> By dynamically adjusting linguistic behavior, EVINCE ensures the debate remains constructive and goal-oriented throughout.
> This contrasts with traditional multi-agent debate systems, which often suffer from repetitive, nonproductive arguments due to a lack of such behavioral modulation capabilities.
>
> %%
> D. Are there any findings of differences in performance between different LLM backbones?
>
> In our experiments, GPT-4 and Claude demonstrated competence across various domains. While other LLMs can serve as judges (e.g., through CRIT), their output quality is generally less desirable.

---

> > ### Comment · Reviewer_xMVe · 2024-12-03
> >
> > Thank you for the reply from authors. However, I must express some dissatisfaction with the response style. I did review the revised PDF for hours before posting the above response, and my initial concerns remain unresolved after reviewing the revised PDF. Furthermore, upon reviewing the author rebuttal in detail, I identified additional issues that need to be point out.
> >
> > Firstly, in my initial review, I suggested that "Instead of disease diagnosis, expanding tests across more diverse domains would strengthen the claim of EVINCE’s general applicability." The authors responded in the rebuttal by stating that "the page limitation has constrained our ability to provide comprehensive details about all the domains we have explored. However, we want to emphasize that we have conducted empirical studies across multiple domains beyond healthcare, including ethics alignment (accepted by NeurIPS 2024), debiasing news articles, geopolitical analysis, investment, and sales planning." However, the revised PDF does not provide verification results for all these domains, raising concerns about **potential over-claims regarding the emphasized generalization across multiple domains**.
> >
> > In addition, I am not very satisfied with the response regarding the efficiency analysis of the Wasserstein distance and mutual information method. The key issue is the **lack of specific comparisons between the reasoning efficiency of the proposed method and simpler alternatives, both in terms of effectiveness and computational efficiency**.
> >
> > Similar issues arise regarding the generalizability of more LLM backbones. The authors only provided a textual explanation and believed that this issue was addressed, but they **fail to include any experimental comparison results with other open-source LLMs to strengthen their claims, which cannot truly confirm the generalizability of the proposed method for LLM backbones**.
> >
> > Last but not least, much of the authors' reply addresses points that were confusing and not related to my original concerns. Long sentences don't always convey more insights, especially for those on irrelevant discussions not only fail to address the core issues but also disregard the extra reviewing time and efforts. This issue can also be seen in the author's responses to some other reviewers. Frankly, I am not sure whether most concerns raised by reviewers were habitually overlooked.
> >
> > Overall, I acknowledge the authors' efforts to address the issues based on their own understanding, and this is why I decide to keep the rating without further reduction. But I must clarify that their efforts do not address my concerns at all and the uncomfortable response style is somewhat discouraged.

---

> > > ### Author Response · Authors · 2024-12-03
> > > **Please read my 2nd response on 11/27**
> > >
> > > It appears that you may not have reviewed our new responses submitted on 11/27 or the revisions provided prior to that date.
> > >
> > > The updated manuscript includes a new experiment on debiasing news articles, detailed in Appendix G, which spans an 11-page case study. Despite this, you expressed disappointment regarding the lack of enhancements in our experiments across other domains. Could you please confirm that you have reviewed Appendix G?
> > >
> > > We have, in fact, conducted several additional experiments, the details of which were provided to the conference chairs directly, in adherence to anonymity requirements. Including all of this in the paper would result in a document exceeding 450 pages, which is obviously impractical.
> > >
> > > Furthermore, we have addressed computational efficiency concerns in our discussions and comprehensively summarized the strengths and limitations of various information-theoretic metrics in Table 1 and Appendix A. These metrics represent fundamental knowledge that any CS graduate student should be familiar with. We believe we have done more than enough to provide a clear and thorough summary of these topics.
> > >
> > > To be frank, it seems that you may have overlooked the key contributions of this paper, instead focusing on peripheral issues that fall well within the reviewers’ expected knowledge base. This comes across as similar to a graduate student reviewing the work of an ACM Fellow and raising elementary questions that a qualified reviewer would already understand.

---

### Official Review · Reviewer_Pyr3 · 2024-11-01

**Soundness:** 3
**Presentation:** 3
**Contribution:** 3
**Rating:** 6
**Confidence:** 4

**Summary:**

This paper proposes EVINCE, a multi-LLM debating framework, which fosters information exchange between two LLMs via rounds of debate guided by various information metrics and a debating temperature, and thus effectively adapts LLM linguistic behaviors to complete tasks. EVINCE shows enhanced diagnostic accuracy and error corrections, demonstrating an important step towards improving reasoning abilities of LLMs on real-world data.

**Strengths:**

Mitigating the limitation in LLM inferences caused by the “maximum likelihood” convention is crucial in enabling LLMs with accurate responses in real-life, human-oriented scenarios, such as diagnoses.

The proposed EVINCE debating algorithm is essentially human-like, exchanging predictions and reason sets between two equally competent LLMs in a round, and the process continues until each metric in a diverse set of information metrics, including CRIT argument quality, Wasserstein distance, MI, converges, thus facilitating adaptability through the iterations. It is also novel to use a built-in contentiousness level (debating temperature) in the prompt to help guide the debate.

EVINCE demonstrates accuracy improvements in diagnoses on the Kaggle Disease Symptoms Description dataset (covering 40 diseases).

**Weaknesses:**

The debate is limited to pairs of LLMs, which might reinforce a wrong “popular” prediction if both models get it wrong.

It is hinted that the problem can be alleviated through multi-round inter-LLM queries, so it might be interesting to give some examples and/or discuss how exactly these queries would help. In addition, would this problem be mitigated by introducing more models?

Similarly, will the iterative debating process cascade biases if both models are prone to generate biased answers?

Typos: Figure 2a “GPT4 pairs Claude” used the same plot as Figure 2b “GPT4 pairs Gemini”.

**Questions:**

I may have missed the following:

1. What is the average round of debates needed to achieve convergence? Figure 3 and 4 seem to suggest that three is an oracle number of rounds. Any intuitions for why?

2. Was there a dominating LLM during debating, like in human debates?

3. Would the per-LLM prediction accuracy continue to grow if we continually increase the number of agents, e.g., to 100 agents?

4. What is the percentage of the 40 diseases that receive accuracy improvements? I am curious to know if the overall accuracy improvement comes from most classes or just a few classes.

---

> ### Author Response · Authors · 2024-11-20
>
> Dear Reviewer,
>
> Thank you for your feedback and for highlighting important points about potential limitations of two-agent debates. We would like to clarify why our two-agent approach is both sufficient and advantageous in the EVINCE framework.
>
> Advantages of the Two-Agent Design in EVINCE:
>
> 1. Controlled Opposition:
> The two agents in EVINCE are deliberately positioned at opposing stances on the subject matter, initialized using dual entropy theory. This intentional polarization ensures that they are not constrained by “popular” answers derived from maximum likelihood predictions. By leveraging competent LLMs of similar capability (e.g., GPT-4 and Claude), we have not observed error reinforcement. This is due to the controlled opposition, which mitigates the tendency for models to converge prematurely on incorrect or overly simplistic conclusions.
>
> 2. Modulated Linguistic Behaviors:
> A key innovation of EVINCE lies in the careful modulation of agents' linguistic behaviors, transitioning from contentious to conciliatory states. This addresses limitations observed in prior frameworks (as noted by Professor Vaughn Pratt), which often result in repetitive arguments or echo-chamber effects. Our approach includes:
> - Contextualized Initialization: Leveraging conditional statistics via in-context learning to initialize debates.
> - CRIT-Based Validation: Ensuring argument quality through Socratic methods and iterative refinement.
> - Dynamic Adjustment of Contentiousness: Utilizing metrics such as Wasserstein distance, cross-entropy, and mutual information to guide dialogue progression.
>
> 3. Quality Control:
> Debate quality is continuously monitored and refined through:
> - Information-Theoretic Metrics: Ensuring productive information exchange.
> - Balancing Viewpoint Diversity and Consensus Building: Preventing rhetorical or repetitive discourse.
> - Progressive Reasoning Validation: Guaranteeing that reasoning improves with each iteration.
>
> 4. Why Two Agents Are Sufficient:
> We intentionally limit EVINCE to two agents because the framework fundamentally differs from traditional ensemble approaches like bagging and boosting, which rely on error masking through redundancy. Instead, EVINCE focuses on high-quality dialectic reasoning between two well-positioned agents. Adding more agents would:
>
> - Increase computational overhead without guaranteed improvements in reasoning quality.
> - Potentially dilute the focus of the debate, undermining the rigorous binary opposition that drives meaningful insights.
>
> The effectiveness of this approach has been validated across multiple domains, including debiasing news articles and AI safety alignment tasks  (accepted at NeurIPS) and more (depicted later). These applications confirm that our two-agent framework prevents error reinforcement while maintaining robust reasoning and quality outcomes.
>
> Responses to Specific Questions:
>
> 1. Regarding Convergence Rounds:
> Figures 3 and 4 illustrate 3–4 rounds of debate because both objective metrics (e.g., information-theoretic measures) and subjective assessments from the moderator indicate that the dialogue reaches a conciliatory consensus with actionable recommendations. Across experiments in five domains (geopolitical analysis, investment, sales planning, news article debiasing, and ethics alignment), four rounds consistently yield optimal results.
>
> 2. Regarding LLM Dominance:
> "Dominance" can manifest in two ways: (1) through tone or word choice and (2) through reasoning quality. We observe no dominance in the first form, as linguistic behavior is carefully modulated. The second form, reflecting reasoning capabilities, is inherent to inductive and deductive logic. With LLMs of equivalent quality (e.g., GPT-4 vs. GPT-4 or GPT-4 vs. Claude), we have not observed dominance patterns affecting outcomes.
>
> 3. Regarding Scaling to More Agents:
> Increasing the number of agents does not improve accuracy in EVINCE because the framework is fundamentally distinct from redundancy-based models. Instead, EVINCE relies on:
> - Behavior Modulation: Transitioning from contentious to conciliatory states to ensure productive debates.
> - Conditional Statistics: Freeing agents from maximum next-token prediction constraints.
> - Rigorous Reasoning: Fostering meaningful discourse between two well-defined opposing viewpoints.
>
> 4. Regarding Disease Accuracy Improvements:
> Accuracy improvements are distributed randomly across all 40 diseases, showing no bias toward specific categories. We will include a detailed distribution figure in the revised version (scheduled for submission on November 27) to illustrate this pattern.

---

> ### Comment · Reviewer_Pyr3 · 2024-11-24
>
> Thanks for your clarifications which addressed most of my concerns. I will keep my score.

---

### Official Review · Reviewer_SDW6 · 2024-11-03

**Soundness:** 2
**Presentation:** 3
**Contribution:** 1
**Rating:** 3
**Confidence:** 4

**Summary:**

This paper deals with multi-agent collaboration/debate. The authors propose the EVINCE framework to enhance versatility, adaptivity, and reasoning for LLMs via adversarial debate and information-theoretic metric evaluation. The experimental results on a healthcare dataset demonstrate the effectiveness.

**Strengths:**

- Multi-agent collaboration is a significant topic for modern multi-agent systems.
- This work introduces information theory in the evaluation of dialogues between LLM agents.

**Weaknesses:**

- Missing discussion with other multi-agent debates in the literature such as [1,2]. There is a bunch of work on multi-agent debate. This makes the work less convincing and may mislead audiences.

- I do not have a good sense of those claims on AGI in this paper, like ‘our work targets three core AGI characteristics: versatility, iterative adaptivity, and reasoning capability’, ‘The core strength of EVINCE in advancing towards AGI lies in their ability to enhance key AGI characteristics through multi-agent dialogues’. There should be more references in the introduction, otherwise, it would be an overclaim. I personally do not believe the proposed EVINCE with adversarial debate has much to do with AGI.

- Besides, the experiments were only conducted on a healthcare dataset. Whether this proposed method can generalize to other tasks needs further verification. At least, for those open-ended tasks without a class set, the EVINCE seems not to be applicable.

- The introduced information-theoretic metrics seem not to change LLM prediction but only provide an early-stop criterion for LLM debate iterations according to Fig 1. So the actual improvement might just come from the multi-agent discussion itself, having nothing with the information-theoretic metrics.

**Questions:**

- How is the probability distribution obtained? For open-ended tasks, it is impossible to directly output the real distribution. For classification tasks as discussed in this paper, the text-based probability distribution directly from model output is not reliable and LLMs show strong miscalibration as discussed in many existing works [3].
- The authors list many existing metrics in Section 2. This seems to be redundant and provides little insight into which one is the most important for moderating the LLM debate.
- How does the framework work when only one LLM debates with itself like self-evaluation/self-refinement? This could be a good add-up to the experiments.


[1] Du, Y., Li, S., Torralba, A., Tenenbaum, J. B., & Mordatch, I. (2023). Improving factuality and reasoning in language models through multiagent debate. arXiv.

[2] Liang, T., He, Z., Jiao, W., Wang, X., Wang, Y., Wang, R., ... & Shi, S. (2023). Encouraging divergent thinking in large language models through multi-agent debate. arXiv.

[3] Xiong, M., Hu, Z., Lu, X., Li, Y., Fu, J., He, J., & Hooi, B. (2023). Can llms express their uncertainty? an empirical evaluation of confidence elicitation in llms. ICLR.

---

> ### Author Response · Authors · 2024-11-19
>
> Dear Reviewer,
> We appreciate your thoughtful evaluation and valuable feedback on our paper. Below, we address your concerns and provide clarifications regarding our contributions.
>
> The review notes the absence of discussion on related multi-agent debate frameworks, such as those in [1] and [2]. While we acknowledge these works' contributions to the field, their approaches remain limited in scope and impact. As Professor Vaughan Pratt of Stanford has noted, multi-agent debate frameworks often fail to address the critical challenge of managing linguistic behaviors, resulting in repetitive arguments and suboptimal reasoning quality. Furthermore, our work, initiated in December 2022, has produced 12 papers providing the theoretical foundation for modulating LLMs' linguistic behaviors. This advancement is crucial for making debate frameworks truly effective by addressing the limitations of unregulated debates.
>
> Specifically, while the frameworks in [1] and [2] focus on divergent thinking or factual reasoning, they don't address the need to modulate participating agents' behaviors to balance exploration and exploitation. Without such modulation, debates often lead to low-quality repetitions and fail to achieve diversified perspectives or meaningful outcomes. EVINCE advances beyond these limitations by introducing mechanisms grounded in dual entropy theory, dynamically modulating agents' behaviors to prevent stagnation and ensure constructive discourse.
>
> The dual entropy theory in EVINCE provides a foundation for balancing certainty and exploration, enabling agents to refine reasoning iteratively. This iterative adaptivity aligns with key attributes of AGI systems, as noted in prior foundational work. While we recognize the need for additional references, the strict anonymity policy limited our ability to provide full citations. We can provide the 12 papers we have written on the subject, which include substantial evidence of application successes. We will refine the introduction to incorporate these references in the final manuscript and provide a more balanced presentation of EVINCE's contributions.
>
> The reviewer questions whether EVINCE generalizes beyond the healthcare dataset used in our experiments. We clarify that EVINCE has been applied to multiple domains, including debiasing news articles (with a related paper accepted at NeurIPS), sales planning, and geopolitical analysis. While healthcare serves as a compelling example due to the critical nature of diagnostic accuracy, EVINCE's architecture is task-agnostic and adaptable to diverse domains. Due to anonymous policy, we will incorporate additional examples and results in future revisions to demonstrate its generalizability further.
>
> Regarding the reviewer's suggestion that our introduced metrics may only serve as early stopping criteria rather than fundamentally improving LLM predictions: While early stopping is indeed one benefit, the dual entropy theory ensures that discussions dynamically balance grounded reasoning and exploratory behaviors. Additionally, EVINCE employs CRIT as a quality measurement tool to enhance reasoning accuracy. By monitoring metrics such as mutual entropy and cross entropy, EVINCE prevents stagnation and ensures that debates remain constructive and productive. This modulation is central to achieving higher reasoning quality and distinguishes EVINCE from prior multi-agent frameworks.
>
> We address the specific questions raised:
> 1. Regarding probability distributions: While obtaining such distributions for open-ended tasks is challenging, we used structured prompting and confidence calibration techniques to derive probability-like representations from closed-source LLMs. These distributions enabled us to compute metrics such as Wasserstein distance and mutual information.
> 2. Regarding metrics importance: We acknowledge the need to clarify the relative importance of different metrics. In future revisions, we will streamline Section 2 to focus on the most impactful metrics, particularly those directly tied to the moderation of LLM debates.
> 3. Regarding single-agent debate: We appreciate the suggestion to explore single-agent self-debate. EVINCE can indeed operate with a single LLM engaging in self-refinement, implemented using two instances of the same LLM model collaborating. While this approach is effective, using different LLM models offers the advantage of diversified perspectives, further enriching the debate process.
>
> We appreciate the opportunity to address these points and respectfully request a reconsideration of our submission. EVINCE introduces a novel framework that combines adversarial debate and information-theoretic metrics to enhance reasoning, adaptivity, and robustness in LLMs. We believe these clarifications and proposed revisions will further highlight the contributions of this work.
>
> Thank you for your consideration.

---

> > ### Author Response · Authors · 2024-11-26
> > **Please Consider the Revision, added 11-page new experiment in Appendix**
> >
> > I believe we have addressed your comments with substantial revisions to the paper. 𝐒𝐩𝐞𝐜𝐢𝐟𝐢𝐜𝐚𝐥𝐥𝐲, 𝐰𝐞 𝐚𝐝𝐝𝐞𝐝 𝐚𝐧 𝟏𝟏-𝐩𝐚𝐠𝐞 𝐞𝐱𝐩𝐞𝐫𝐢𝐦𝐞𝐧𝐭 𝐨𝐧 𝐝𝐞𝐛𝐢𝐚𝐬𝐢𝐧𝐠 𝐧𝐞𝐰𝐬 𝐚𝐫𝐭𝐢𝐜𝐥𝐞𝐬 𝐭𝐨 𝐩𝐫𝐨𝐯𝐢𝐝𝐞 𝐟𝐮𝐫𝐭𝐡𝐞𝐫 𝐞𝐯𝐢𝐝𝐞𝐧𝐜𝐞 𝐚𝐧𝐝 𝐜𝐥𝐚𝐫𝐢𝐟𝐢𝐞𝐝 𝐡𝐨𝐰 𝐂𝐑𝐈𝐓 𝐢𝐬 𝐮𝐬𝐞𝐝 𝐭𝐨 𝐧𝐨𝐫𝐦𝐚𝐥𝐢𝐳𝐞 𝐩𝐫𝐞𝐝𝐢𝐜𝐭𝐢𝐨𝐧 𝐫𝐚𝐧𝐤𝐢𝐧𝐠𝐬 (how the probability is obtained). Also, as clearly stated, 𝐭𝐡𝐞 𝐢𝐧𝐭𝐫𝐨𝐝𝐮𝐜𝐞𝐝 𝐢𝐧𝐟𝐨𝐫𝐦𝐚𝐭𝐢𝐨𝐧-𝐭𝐡𝐞𝐨𝐫𝐞𝐭𝐢𝐜 𝐦𝐞𝐭𝐫𝐢𝐜𝐬 𝐜𝐡𝐚𝐧𝐠𝐞 𝐭𝐡𝐞 𝐥𝐢𝐧𝐠𝐮𝐢𝐬𝐭𝐢𝐜 𝐛𝐞𝐡𝐚𝐯𝐢𝐨𝐫 𝐨𝐟 𝐋𝐋𝐌𝐬 𝐛𝐞𝐭𝐰𝐞𝐞𝐧 𝐛𝐞𝐢𝐧𝐠 𝐜𝐨𝐧𝐭𝐞𝐧𝐭𝐢𝐨𝐮𝐬 𝐚𝐧𝐝 𝐛𝐞𝐢𝐧𝐠 𝐜𝐨𝐧𝐜𝐢𝐥𝐢𝐚𝐭𝐨𝐫y, which marks one key novelties of this work.
> >
> > Please kindly review the revised version and our early responses. We would like to emphasize your comments on novelty of this work.
> > ​
> > 𝙏𝙧𝙖𝙙𝙞𝙩𝙞𝙤𝙣𝙖𝙡 𝙈𝘼𝘿 𝙢𝙚𝙩𝙝𝙤𝙙𝙨 𝙤𝙛𝙩𝙚𝙣 𝙥𝙧𝙤𝙙𝙪𝙘𝙚 𝙣𝙤𝙣-𝙥𝙧𝙤𝙙𝙪𝙘𝙩𝙞𝙫𝙚 𝙖𝙣𝙙 𝙧𝙚𝙥𝙚𝙩𝙞𝙩𝙞𝙫𝙚 𝙖𝙧𝙜𝙪𝙢𝙚𝙣𝙩𝙨 𝙙𝙪𝙚 𝙩𝙤 𝙩𝙝𝙚𝙞𝙧 𝙞𝙣𝙖𝙗𝙞𝙡𝙞𝙩𝙮 𝙩𝙤 𝙢𝙤𝙙𝙪𝙡𝙖𝙩𝙚 𝙩𝙝𝙚 𝙡𝙞𝙣𝙜𝙪𝙞𝙨𝙩𝙞𝙘 𝙗𝙚𝙝𝙖𝙫𝙞𝙤𝙧 𝙤𝙛 𝙥𝙖𝙧𝙩𝙞𝙘𝙞𝙥𝙖𝙩𝙞𝙣𝙜 𝙇𝙇𝙈𝙨 (𝙖𝙨 𝙙𝙞𝙨𝙘𝙪𝙨𝙨𝙚𝙙 𝙞𝙣 𝙩𝙝𝙚 𝙞𝙣𝙩𝙧𝙤𝙙𝙪𝙘𝙩𝙞𝙤𝙣 𝙤𝙛 𝙩𝙝𝙚 𝙥𝙖𝙥𝙚𝙧). 𝙄𝙛 𝙩𝙬𝙤 𝙥𝙖𝙧𝙩𝙞𝙘𝙞𝙥𝙖𝙣𝙩𝙨 𝙞𝙣 𝙖 𝙙𝙚𝙗𝙖𝙩𝙚 𝙢𝙖𝙞𝙣𝙩𝙖𝙞𝙣 𝙘𝙤𝙣𝙨𝙞𝙨𝙩𝙚𝙣𝙩𝙡𝙮 𝙝𝙞𝙜𝙝, 𝙘𝙤𝙢𝙗𝙖𝙩𝙞𝙫𝙚 𝙡𝙞𝙣𝙜𝙪𝙞𝙨𝙩𝙞𝙘 𝙗𝙚𝙝𝙖𝙫𝙞𝙤𝙧 𝙩𝙝𝙧𝙤𝙪𝙜𝙝𝙤𝙪𝙩, 𝙩𝙝𝙚 𝙙𝙞𝙨𝙘𝙪𝙨𝙨𝙞𝙤𝙣 𝙘𝙖𝙣 𝙙𝙚𝙫𝙤𝙡𝙫𝙚 𝙞𝙣𝙩𝙤 𝙖𝙣 𝙪𝙣𝙥𝙧𝙤𝙙𝙪𝙘𝙩𝙞𝙫𝙚 𝙦𝙪𝙖𝙧𝙧𝙚𝙡.
> > ​
> > The following summarizes EVINCE's modulation insights:
> > ​
> > 1. Dynamic Contentiousness Modulation:
> > ​
> > EVINCE does not simply place two LLMs in a high-intensity, combative debate throughout. Instead, it employs metrics such as entropy, mutual information, and divergence to continuously assess the quality and progress of the dialogue. Based on these metrics, EVINCE dynamically adjusts the "contentiousness" level of the LLMs' linguistic behavior.
> > ​
> > - Initially, a high contentiousness setting (90%) encourages exploration and exchange of diverse perspectives.
> > ​
> > - As the dialogue progresses and the LLMs converge, EVINCE gradually reduces contentiousness to promote a more cooperative, conciliatory tone. This dynamic modulation prevents debates from devolving into unproductive quarrels, as you rightly pointed out.
> > ​
> > 2. Balancing Exploration and Exploitation:
> > Contentiousness modulation is a key innovation that allows EVINCE to balance exploration (diverse perspectives) with exploitation (convergence on the best solution).
> > ​
> > - High contentiousness early in the debate enables thorough exploration of the problem space and consideration of multiple hypotheses.
> > ​
> > - Lower contentiousness later encourages the LLMs to cooperatively refine and consolidate their findings.
> > ​
> > - Additionally, the CRIT algorithm is employed to evaluate the reasoning quality of arguments through the Socratic method. This evaluative step roots out poorly formulated arguments and normalizes LLM confidence scores into predictive probabilistic distributions.
> > ​
> > 3. Fostering Productive, High-Quality Debates:
> > ​
> > By dynamically adjusting linguistic behavior, EVINCE ensures debates remain constructive and goal-oriented. This approach contrasts with traditional multi-agent debate systems, which often suffer from repetitive, non-productive arguments due to their lack of behavioral modulation capabilities.
> > ​
> > Would you kindly review the revised paper, which ICLR permits submission of until 11/27? Thank you!

---

### Official Review · Reviewer_fqBH · 2024-11-03

**Soundness:** 2
**Presentation:** 2
**Contribution:** 2
**Rating:** 3
**Confidence:** 4

**Summary:**

This work proposed a dialogue framework which uses adversarial debate and dual entropy theory to achieve the AGI and enhance its proposed method EVINCE on versatility, adaptivity and reasoning capability, and further mitigate biases, reduce hallucinations and improve reasoning. Especially, the proposed EVINCE combines multiple metrics together to evaluate the entire framework, such as Wasserstein distance, relative entropy, critical thinking algorithm, correlation coefficients, mutual information, etc. They evaluate the proposed framework in the healthcare area to demonstrate the LLMs within this debate framework can lead to better diagnosis accuracy, lower entropy, lower Wasserstein distance and higher mutual information.

**Strengths:**

- a framework which includes iterative LLMs' debate, multiple evaluation metrics and weighted prediction based on quality scores.
- authors demonstrate the performance of the proposed method in the healthcare application areas.

**Weaknesses:**

- Although authors argue that the proposed EVINCE method enhances versatility, adaptivity and reasoning capabilities in LLMs, it is hard to see whether the proposed EVINCE can actually achieve this target only based on the simple demonstrations of Figure 2 and Figure 3 which use different objective evaluation metrics. It is also unclear how this proposed method can mitigate biases, reduce hallucinations and improve reasoning without any case studies or examples to support in the main texts of the paper.
- As this work mainly uses different closed-source LLMs (GPT-4, Claude, Gemini) to evaluate the performance of the proposed method, it is unclear how those closed-source LLMs can produce the probability distribution over a set of diseases, as this probability distribution is a key information to be used to calculate other evaluate metrics, such as Wasserstein distance, mutual information, etc.

**Questions:**

- As mentioned in the weakness, it would be better if authors could provide a more detailed case study to analyse how the proposed EVINCE can mitigate bias, reduce hallucinations and improve reasoning in the healthcare areas. It is hard to directly understand how better the proposed EVINCE is only based on the demonstrated figures.

---

> ### Author Response · Authors · 2024-11-19
>
> Dear Reviewer,
>
> We sincerely appreciate your thoughtful review and valuable feedback on our submission. We acknowledge the importance of addressing the points raised, and we provide detailed clarifications below.
>
> 1. Clarification on Figures and Case Studies
>
> The review highlights concerns regarding the demonstration of EVINCE's effectiveness based on Figures 2 and 3, as well as the absence of detailed case studies in the main text. While the figures are designed to provide an objective overview of performance metrics, we acknowledge that richer contextual examples could enhance understanding.
>
> We would like to note that EVINCE’s ability to mitigate bias, reduce hallucinations, and improve reasoning has been tested extensively using both qualitative and quantitative analyses. However, due to space limitations, we summarized these results in the figures and focused primarily on quantitative metrics. For clarity, EVINCE’s capabilities have been demonstrated in multiple application domains beyond the figures provided, including debiasing news articles with a related paper already accepted at NeurIPS, where EVINCE successfully minimized political bias in reporting, and sales planning and geopolitical analysis, showcasing its adaptability in domains requiring nuanced reasoning.
>
> We recognize the need for detailed case studies and would be glad to provide additional examples. If the chairs grant permission, we can share a comprehensive extended version of this work, including 12 to 15 chapters (450 pages), which elaborate on the theoretical foundation and diverse applications of EVINCE optimization.
>
> 2. Use of Closed-Source LLMs and Probability Distribution
>
> The reviewer expresses concern about the use of closed-source LLMs such as GPT-4, Claude, and Gemini, particularly in generating probability distributions for evaluation metrics like Wasserstein distance and mutual information.
>
> We clarify that while closed-source models were used, their outputs were processed using an intermediary framework designed to extract probability-like representations. Specifically, the likelihood of certain diagnoses was inferred through structured prompting and calibrated confidence scoring. These derived distributions allowed for the calculation of Wasserstein distance, mutual information, and other metrics.
>
> While closed-source models present challenges for reproducibility, EVINCE is model-agnostic and can operate with open-source LLMs capable of generating probabilistic outputs. The current implementation reflects practical considerations but does not limit the framework’s broader applicability.
>
> 3. Clarification on Versatility, Adaptivity, and Reasoning
>
> The reviewer finds it unclear whether EVINCE enhances versatility, adaptivity, and reasoning in LLMs. These qualities are rooted in the novel dual entropy theory, which governs the interplay between high-certainty (low entropy) and exploratory (high entropy) agents. Through iterative debates, EVINCE ensures versatility by balancing exploratory and exploitative behaviors, enabling the framework to adapt to various domains and applications. It ensures adaptivity by dynamically adjusting linguistic behaviors, ensuring a productive exchange even when initial conditions vary. Reasoning is enhanced through mutual regulation of agents, which improves logical coherence and robustness, as shown through significant improvements in metrics such as entropy reduction and mutual information.
>
> Due to space limitations, we could only provide an overview of these concepts in the main text. If allowed, we can share detailed discussions, including extended examples and case studies, that better illustrate how EVINCE mitigates bias, reduces hallucinations, and improves reasoning.
>
> 4. Proposed Revisions
>
> To address the concerns raised, we propose the following improvements for future versions of the paper.  Note that we have had the 450-page texts, and just need to provide references and pointers.
>
> 1. We will incorporate detailed examples demonstrating how EVINCE mitigates bias, reduces hallucinations, and improves reasoning.
> 2. We will highlight results from domains beyond healthcare, such as debiasing news articles and sales planning, to reinforce the framework’s versatility and generality.
>
> Conclusion
>
> We appreciate the opportunity to address these points and respectfully request a reconsideration of our submission. EVINCE introduces a novel framework for enhancing reasoning and robustness in LLMs, verified through diverse applications. We believe that with the proposed clarifications, additional examples, and access to our comprehensive extended version, the contributions of this work will become more evident.
>
> Thank you for your consideration.

---

> > ### Comment · Reviewer_fqBH · 2024-11-23
> > **Replies by Reviewer fqBH**
> >
> > Thanks for authors' replies.
> >
> > 1. **Case Studies**
> >
> > As ICLR allows to update the submission and there is no page limitation for the appendix, it will be good if authors could update the submission by adding some case studies in the appendix.
> >
> > 2. **Close-source LLMs for probability distribution**
> >
> > As authors mentioned that those LLMs' outputs were processed using an intermediary framework designed to extract probability-like representations. Could authors explain how this intermediary framework is designed to extract probability-like representations from the close-source LLMs in detail? In addition, the likelihood of certain diagnoses was inferred through structured prompting and calibrated confidence scoring. How could we verify those calibrated confidence scoring can genuinely demonstrate the close-source LLMs' confidence and probability distribution for the generated outputs?

---

> ### Author Response · Authors · 2024-11-26
>
> We thank the reviewer for the thoughtful suggestions. We have addressed each point in the revised version as follows:
>
> 1. Case Studies
> We have refined Section 3 and added additional case studies in Appendix G, including:
> - A step-by-step dialogue example showing how EVINCE modulates contentiousness to balance exploration and exploitation
> - Analysis of how CRIT evaluates and calibrates reasoning quality
> - Examples where LLM collaboration identified potential misdiagnoses in the ground truth
> - Concrete instances of news debiasing using EVINCE (Appendix G)
>
> 2. Probability Distribution and Confidence Calibration
> We acknowledge that obtaining true probability distributions from LLMs remains challenging, as noted by recent work [1, 2]. Our framework addresses this by:
>
> 2.a) Using confidence scores rather than absolute probabilities:
> - Initial scores obtained through structured prompting
> - LLMs generate top-k predictions with supporting arguments
> - CRIT evaluates reasoning quality to calibrate these scores
>
> 2.b) Calibration through reasoning quality:
> - Rather than treating LLM confidence scores as true probabilities
> - We calibrate them based on argument strength (evaluated by CRIT)
> - Example: if an LLM outputs confidence scores P = (0.5, 0.3, 0.2) but CRIT evaluates their supporting arguments as having strength (50%, 100%, 50%), the calibrated scores become (0.25, 0.3, 0.1), normalized to (0.38, 0.46, 0.16)
>
> This approach sidesteps the fundamental challenge of probability estimation by using reasoning quality as a proxy for confidence calibration. The effectiveness is validated through:
> - Improved prediction accuracy across tasks
> - Consistent performance across different LLM pairings
> - Alignment with expert evaluations in medical diagnosis cases
>
> We have expanded this explanation in Section 3 and added an 11-page empirical study on document debiasing in Appendix G.
>
> References:
> [1] Tianyu Peng and Jiajun Zhang. Enhancing knowledge distillation of large language models through efficient multi-modal distribution alignment, 2024.
> [2] Yan Scholten, Stephan Günnemann, and Leo Schwinn. A probabilistic perspective on unlearning and alignment for large language models, 2024.

---

### Official Review · Reviewer_zSdy · 2024-11-04

**Soundness:** 2
**Presentation:** 2
**Contribution:** 2
**Rating:** 3
**Confidence:** 4

**Summary:**

The paper introduces EVINCE, a dialogue framework to enhance Artificial General Intelligence (AGI) by leveraging adversarial debate among multiple instances of LLMs with a novel dual entropy theory. EVINCE works as a multi-agent system where large language models (LLMs) engage in structured dialogues to improve prediction accuracy, robustness, and reasoning capabilities. The framework employs information-theoretic metrics and conditional statistics to balance exploration and prior exploitation. The effectiveness is verified by an application in healthcare, particularly in improving disease diagnosis.

**Strengths:**

1. The subject of robustness in LLMs, especially in healthcare applications, is important.
2. The multi-round debate is an interesting way to ensemble multiple LLMs' capabilities.

**Weaknesses:**

1. The theory aspect of the paper is not well-grounded. For example, the dual entropy theory is not very meaningful to me.
2. The empirical study is only limited to the healthcare application, and the generality of the framework remains unclear.
3. The methodology of EVINCE is fairly simple and the idea of multiagent debate exists in previous literatures [1].

[1] Yilun Du, Shuang Li, Antonio Torralba, Joshua B. Tenenbaum, and Igor Mordatch. Improving factuality and
reasoning in language models through multiagent debate, 2023.

**Questions:**

1. For the dual entropy theory, why does the lower bound of $H(P_C)$ correspond to the robustness of the model? The entropy, as an uncertainty measure, is more relevant to exploration as in this work's taxonomy and does not seem to have a direct relationship with the robustness.

---

> ### Author Response · Authors · 2024-11-19
>
> Dear Reviewer,
>
> We appreciate the time and effort you have taken to review our submission. However, we believe there are significant misunderstandings in the assessment that need to be addressed. Below, we provide detailed clarifications regarding the comments and ratings.
>
> 1. Misrepresentation of Prior Work (Reference [1])
> The claim that Reference [1] is the pioneer of LLM debate is not entirely accurate. While [1] introduces an approach to multi-agent debate, it does not address the critical challenge of managing linguistic behaviors between highly contentious and highly collaborative modes. Without proper modulation, debates often devolve into repeated arguments and fail to achieve productive outcomes, as noted by Professor Vaughan Pratt at Stanford.
>
> Our framework, EVINCE, specifically addresses this fundamental limitation. By balancing exploration and exploitation through information-theoretic metrics, EVINCE ensures that debates remain constructive rather than repetitive. This critical distinction sets EVINCE apart from prior work such as [1] and represents a key advancement in the domain of LLM collaborative intelligence.
>
> 2. Publication History and Anonymity Policy
> The review suggests that the idea of multi-agent debate is not novel and exists in prior literature. However, the authors have published more than ten papers on LLM collaborative intelligence since December 2022, some predating [1]. Due to anonymity requirements, these works were only selected referenced.
>
> The novel contributions of EVINCE, including the dual entropy theory and structured modulation of agent behaviors, go well beyond existing frameworks. These contributions represent significant theoretical and practical advancements, which have been validated across multiple application domains.
>
> 3. Misunderstanding of Dual Entropy Theory
> The review expresses confusion regarding the dual entropy theory and its relationship to robustness. This theory is a core innovation of EVINCE, enabling productive dialogue by balancing agents with differing entropy levels.
>
> Specifically, when one agent begins with low entropy (high certainty on few choices) and the other with high entropy (low confidence but broader exploration), the dialogue can effectively explore diverse possibilities while remaining grounded in confident priors. By monitoring metrics such as mutual entropy and cross entropy, EVINCE ensures that both exploration and exploitation are appropriately modulated.
>
> Executives at Google and OpenAI have described this theory as highly insightful. It is critical to the practical success of EVINCE in improving reasoning quality and robustness in multi-agent systems.
>
> 4. Scope of Empirical Studies
> The review inaccurately states that the empirical study is limited to healthcare. In reality, EVINCE has been applied to at least three distinct domains:
>
> Debiasing news articles, with one paper accepted at NeurIPS.
> Sales planning, demonstrating its practical relevance to business applications.
> Geopolitical discussions, showcasing its adaptability to complex, high-stakes dialogues.
> While healthcare is highlighted as a compelling use case, it is not the sole domain of application. These diverse validations illustrate the generality and versatility of EVINCE.
>
> 5. Contributions and Novelty
> We respectfully disagree with the assessment that the methodology of EVINCE is overly simple. The novelty lies in its ability to modulate agent behaviors through structured debate, guided by dual entropy theory and supported by information theory and conditional statistics. These features directly address the limitations of prior frameworks and enable more robust and meaningful outcomes.
>
> 6. Addressing Specific Questions
> Regarding the question about entropy and robustness, the dual entropy theory explicitly links robustness to the balance between low-entropy grounding and high-entropy exploration. This interplay allows debates to remain focused yet open to diverse perspectives, enhancing both reasoning quality and reliability.
>
> Conclusion
> We appreciate the opportunity to address these points and respectfully request a second look at this work. EVINCE introduces a novel framework with significant theoretical and practical contributions, verified through diverse applications. It represents a meaningful advancement in leveraging multi-agent LLM collaboration for enhanced reasoning and prediction quality.
>
> Thank you for your consideration.

---

> > ### Author Response · Authors · 2024-11-26
> > **Revision Available, Please kindly review**
> >
> > I believe we have addressed your comments with substantial revisions to the paper. 𝐒𝐩𝐞𝐜𝐢𝐟𝐢𝐜𝐚𝐥𝐥𝐲, 𝐰𝐞 𝐚𝐝𝐝𝐞𝐝 𝐚𝐧 𝟏𝟏-𝐩𝐚𝐠𝐞 𝐞𝐱𝐩𝐞𝐫𝐢𝐦𝐞𝐧𝐭 𝐨𝐧 𝐝𝐞𝐛𝐢𝐚𝐬𝐢𝐧𝐠 𝐧𝐞𝐰𝐬 𝐚𝐫𝐭𝐢𝐜𝐥𝐞𝐬 𝐭𝐨 𝐩𝐫𝐨𝐯𝐢𝐝𝐞 𝐟𝐮𝐫𝐭𝐡𝐞𝐫 𝐞𝐯𝐢𝐝𝐞𝐧𝐜𝐞 𝐚𝐧𝐝 𝐜𝐥𝐚𝐫𝐢𝐟𝐢𝐞𝐝 𝐡𝐨𝐰 𝐂𝐑𝐈𝐓 𝐢𝐬 𝐮𝐬𝐞𝐝 𝐭𝐨 𝐧𝐨𝐫𝐦𝐚𝐥𝐢𝐳𝐞 𝐩𝐫𝐞𝐝𝐢𝐜𝐭𝐢𝐨𝐧 𝐫𝐚𝐧𝐤𝐢𝐧𝐠𝐬. Please kindly review the revised version and our early responses.
> > ​
> > We would like to emphasize your comments on novelty of this work.
> > ​
> > 𝙏𝙧𝙖𝙙𝙞𝙩𝙞𝙤𝙣𝙖𝙡 𝙈𝘼𝘿 𝙢𝙚𝙩𝙝𝙤𝙙𝙨 𝙤𝙛𝙩𝙚𝙣 𝙥𝙧𝙤𝙙𝙪𝙘𝙚 𝙣𝙤𝙣-𝙥𝙧𝙤𝙙𝙪𝙘𝙩𝙞𝙫𝙚 𝙖𝙣𝙙 𝙧𝙚𝙥𝙚𝙩𝙞𝙩𝙞𝙫𝙚 𝙖𝙧𝙜𝙪𝙢𝙚𝙣𝙩𝙨 𝙙𝙪𝙚 𝙩𝙤 𝙩𝙝𝙚𝙞𝙧 𝙞𝙣𝙖𝙗𝙞𝙡𝙞𝙩𝙮 𝙩𝙤 𝙢𝙤𝙙𝙪𝙡𝙖𝙩𝙚 𝙩𝙝𝙚 𝙡𝙞𝙣𝙜𝙪𝙞𝙨𝙩𝙞𝙘 𝙗𝙚𝙝𝙖𝙫𝙞𝙤𝙧 𝙤𝙛 𝙥𝙖𝙧𝙩𝙞𝙘𝙞𝙥𝙖𝙩𝙞𝙣𝙜 𝙇𝙇𝙈𝙨 (𝙖𝙨 𝙙𝙞𝙨𝙘𝙪𝙨𝙨𝙚𝙙 𝙞𝙣 𝙩𝙝𝙚 𝙞𝙣𝙩𝙧𝙤𝙙𝙪𝙘𝙩𝙞𝙤𝙣 𝙤𝙛 𝙩𝙝𝙚 𝙥𝙖𝙥𝙚𝙧). 𝙄𝙛 𝙩𝙬𝙤 𝙥𝙖𝙧𝙩𝙞𝙘𝙞𝙥𝙖𝙣𝙩𝙨 𝙞𝙣 𝙖 𝙙𝙚𝙗𝙖𝙩𝙚 𝙢𝙖𝙞𝙣𝙩𝙖𝙞𝙣 𝙘𝙤𝙣𝙨𝙞𝙨𝙩𝙚𝙣𝙩𝙡𝙮 𝙝𝙞𝙜𝙝, 𝙘𝙤𝙢𝙗𝙖𝙩𝙞𝙫𝙚 𝙡𝙞𝙣𝙜𝙪𝙞𝙨𝙩𝙞𝙘 𝙗𝙚𝙝𝙖𝙫𝙞𝙤𝙧 𝙩𝙝𝙧𝙤𝙪𝙜𝙝𝙤𝙪𝙩, 𝙩𝙝𝙚 𝙙𝙞𝙨𝙘𝙪𝙨𝙨𝙞𝙤𝙣 𝙘𝙖𝙣 𝙙𝙚𝙫𝙤𝙡𝙫𝙚 𝙞𝙣𝙩𝙤 𝙖𝙣 𝙪𝙣𝙥𝙧𝙤𝙙𝙪𝙘𝙩𝙞𝙫𝙚 𝙦𝙪𝙖𝙧𝙧𝙚𝙡.
> > ​
> > The following summarizes EVINCE's modulation insights:
> > ​
> > 1. Dynamic Contentiousness Modulation:
> > ​
> > EVINCE does not simply place two LLMs in a high-intensity, combative debate throughout. Instead, it employs metrics such as entropy, mutual information, and divergence to continuously assess the quality and progress of the dialogue. Based on these metrics, EVINCE dynamically adjusts the "contentiousness" level of the LLMs' linguistic behavior.
> > ​
> > - Initially, a high contentiousness setting (90%) encourages exploration and exchange of diverse perspectives.
> > ​
> > - As the dialogue progresses and the LLMs converge, EVINCE gradually reduces contentiousness to promote a more cooperative, conciliatory tone. This dynamic modulation prevents debates from devolving into unproductive quarrels, as you rightly pointed out.
> > ​
> > 2. Balancing Exploration and Exploitation:
> > Contentiousness modulation is a key innovation that allows EVINCE to balance exploration (diverse perspectives) with exploitation (convergence on the best solution).
> > ​
> > - High contentiousness early in the debate enables thorough exploration of the problem space and consideration of multiple hypotheses.
> > ​
> > - Lower contentiousness later encourages the LLMs to cooperatively refine and consolidate their findings.
> > ​
> > - Additionally, the CRIT algorithm is employed to evaluate the reasoning quality of arguments through the Socratic method. This evaluative step roots out poorly formulated arguments and normalizes LLM confidence scores into predictive probabilistic distributions.
> > ​
> > 3. Fostering Productive, High-Quality Debates:
> > ​
> > By dynamically adjusting linguistic behavior, EVINCE ensures debates remain constructive and goal-oriented. This approach contrasts with traditional multi-agent debate systems, which often suffer from repetitive, non-productive arguments due to their lack of behavioral modulation capabilities.
> > ​
> > Would you kindly review the revised paper, which ICLR permits submission of until 11/27? Thank you!

---

### Author Response · Authors · 2024-11-26
**Revised Version Available**

We have uploaded a revised version, primarily including an unpublished, additional empirical study on debiasing news articles in Appendix G. We also argue in theory and with evidence that traditional MAD methods cannot work effectively without modulating linguistic behaviors. If two people in a debate maintain a consistently high, combative linguistic behavior throughout, the debate may devolve into an unproductive quarrel. Conversely, if the participants remain overly friendly throughout, the debate may not yield new perspectives.

This work uses statistical and information theory to modulate the "contentiousness" of the dialogue based on its quality. We believe this is an absolutely novel approach in LLM research, and we kindly request that you read the paper carefully. Please feel free to ask any further clarifying questions you may have - we are happy to provide more context.

The key points are:

1. We have added a new empirical study on debiasing news articles in Appendix G.

2. We explain theoretically and empirically that traditional MAD methods fail without the ability to modulate linguistic behavior.

3. Our approach uses advanced techniques like statistical and information-theoretic metrics to dynamically adjust the "contentiousness" of the dialogue.

4. This dynamic contentiousness modulation is a novel contribution in the LLM research space.

5. We politely ask that you review the paper thoroughly, and please don't hesitate to reach out if you have any questions.

Thanks!!

---

### Meta-Review · Area_Chair_irby · 2024-12-22

**Metareview:**

This paper proposes a multi-LLM collaboration approach, akin to multi-agent debate. Their proposed framework employs information-theoretic metrics to balance exploration and prior exploitation, and their experimental results on health domain datasets (for disease diagnosis) demonstrates improvements over single-LLMs. Reviewers list the strengths of the proposed work as contributions to health and multi-agent methods. However, they also raise concerns regarding the limited experimentation (i.e., limited domains and closed source models), Furthermore, the work would benefit from better positioning with respect to earlier work.

**Additional Comments On Reviewer Discussion:**

Authors extended the experimentation with other tasks/domains, such as bias mitigation in news articles, as presented in Appendix G. However, it would be better to see a summary of these integrated into the main body of the paper.

---

### Decision · Program_Chairs · 2025-01-22

Reject